EMBO
Molecular Medicine

# A genome-wide RNAi screen reveals essential therapeutic targets of breast cancer stem cells

Abir Arfaoui[1,2,3], Claire Rioualen[4], Violette Azzoni[1,†], Guillaume Pinna[5,†], Pascal Finetti[6], Julien Wicinski[1], Emmanuelle Josselin[7], Manon Macario[1], Rémy Castellano[7], Candi Léonard-Stumpf[1], Anthony Bal[1], Abigaelle Gros[1], Sylvain Lossy[5], Maher Kharrat[2], Yves Collette[7], Francois Bertucci[6], Daniel Birnbaum[6], Hayet Douik[2,3], Ghislain Bidaut[4], Emmanuelle Charafe-Jauffret[1] & Christophe Ginestier[1,*] 🆔

## Abstract

Therapeutic resistance is a major clinical challenge in oncology. Evidence identifies cancer stem cells (CSCs) as a driver of tumor evolution. Accordingly, the key stemness property unique to CSCs may represent a reservoir of therapeutic target to improve cancer treatment. Here, we carried out a genome-wide RNA interference screen to identify genes that regulate breast CSCs-fate (bCSC). Using an interactome/regulome analysis, we integrated screen results in a functional mapping of the CSC-related processes. This network analysis uncovered potential therapeutic targets controlling bCSC-fate. We tested a panel of 15 compounds targeting these regulators. We showed that mifepristone, salinomycin, and JQ1 represent the best anti-bCSC activity. A combination assay revealed a synergistic interaction of salinomycin/JQ1 association to deplete the bCSC population. Treatment of primary breast cancer xenografts with this combination reduced the tumor-initiating cell population and limited metastatic development. The clinical relevance of our findings was reinforced by an association between the expression of the bCSC-related networks and patient prognosis. Targeting bCSCs with salinomycin/JQ1 combination provides the basis for a new therapeutic approach in the treatment of breast cancer.

**Keywords** breast cancer; cancer stem cells; JQ1; RNAi screen; salinomycin
**Subject Categories** Cancer; Chemical Biology; Regenerative Medicine

## Introduction

Our understanding of cancer is being transformed by exploring clonal diversity, cell plasticity, drug resistance, and causation within an evolutionary framework. It is now well accepted that neoplasms change over time through an evolutionary process at the cell level, driven by genetic and epigenetic alterations, and especially under sustained treatment (Maley *et al*, 2017). This evolution explains the processes of tumor progression and therapeutic resistance. While tumor cell heterogeneity composes a complex ecosystem, cancer stem cells (CSCs) seem to orchestrate the evolutionary selection in cancer (Kreso & Dick, 2014; Greaves, 2015). A direct consequence is the pivotal role of CSCs in leading clinical evolution. We and others have clearly established a positive association between CSCs burden and disease progression or therapeutic failure (Ginestier *et al*, 2007; Eppert *et al*, 2011; Merlos-Suarez *et al*, 2011; Charafe-Jauffret *et al*, 2013). Thus, developing CSC-targeting therapies is of major interest and requires insight into the underlying mechanisms regulating stemness properties. While CSC self-renewal capability offers to tumors a unique capacity to promote cancer recurrence, it may also unveil therapeutic opportunities to control tumor progression through clinically relevant methods targeting CSC dynamics. Previous studies have reported strong evidence supporting the concept of CSC control as a therapeutic strategy. In colorectal cancer, targeting the BMI1-related self-renewal machinery, using a small-molecule inhibitor (PTC-209), irreversibly reduces the potential of CSCs to initiate tumors (Kreso *et al*, 2014). BBI608, a potent small-molecule inhibitor of STAT3, efficiently targets pancreatic CSCs, suppressing tumor relapse and blocking metastasis formation (Li *et al*, 2015). We reported similar observations in breast cancer using repertaxin, a CXCR1 inhibitor. Repertaxin treatment was able to specifically

1 Inserm, CNRS, Institut Paoli-Calmettes, CRCM, Epithelial Stem Cells and Cancer Lab, Aix-Marseille Univ, Marseille, France
2 Faculté de Médecine de Tunis, LR99ES10 Laboratoire de Génétique Humaine, Université de Tunis El Manar, Tunis, Tunisia
3 Service de Biologie Clinique, Institut Salah Azaiz, Tunis, Tunisia
4 Inserm, CNRS, Institut Paoli-Calmettes, CRCM, Plateform Integrative Bioinformatics, Cibi, Aix-Marseille Univ, Marseille, France
5 Plateforme ARN Interférence, Service de Biologie Intégrative et de Génétique Moléculaire (SBIGeM), I2BC, CEA, CNRS, Université Paris-Saclay, Gif-sur-Yvette, France
6 Inserm, CNRS, Institut Paoli-Calmettes, CRCM, Molecular Oncology "Equipe labellisée Ligue Contre le Cancer", Aix-Marseille Univ, Marseille, France
7 Inserm, CNRS, Institut Paoli-Calmettes, CRCM, TrGET Plateform, Aix-Marseille Univ, Marseille, France
*Corresponding author. Tel: +33 4 91 22 35 09; Fax: +33 4 91 22 35 44; E-mail: christophe.ginestier@inserm.fr
†These authors contributed equally to this work

target the breast CSC (bCSC) population in patient-derived xenografts (PDXs), delaying tumor growth and reducing metastasis spread (Ginestier *et al*, 2010). A new generation of clinical trials is currently enrolling patients to validate CSC self-renewal process as a clinically relevant therapeutic target (Kaiser, 2015). The proposition of additional therapeutics targeting CSC self-renewal is mandatory to strengthen the use of self-renewing cells as the primary targetable entity to improve cancer treatment.

In this study, we proposed a new strategy to identify a set of novel essential functional regulator genes of bCSCs, with potential as therapeutic targets. We have recently developed a high-content analysis strategy to perform gain- or loss-of-function screens with bCSCs-fate as a readout (El Helou *et al*, 2017). This strategy allowed the identification of switcher miRNAs that regulate bCSC-fate and tumor progression in a bimodal fashion. Here, we have conducted a genetic screen using a human genome-wide RNAi library to identify intrinsic molecular pathways that regulate bCSC self-renewal. Our work uncovered specific gene subnetworks enriched in regulator genes of bCSCs with potential as therapeutic targets. We selected a panel of compounds developed for targeted therapies to interfere with the different gene subnetworks and identified three potent drugs able to reduce the bCSC pool *in vitro*. We next validated in PDXs the best drug combinations to eradicate the bCSC population. We also described a positive association between the expression levels of genes from the self-renewal subnetworks and the patient outcome, reinforcing their clinical relevance in breast cancer therapy. Thus, our data strongly support that inhibiting regulator genes of bCSC self-renewal could be an effective approach to control tumor growth. This strategy opens new therapeutic perspectives in cancer treatment, with potential applications in clinics.

# Results

## Genome-wide RNAi screen to identify regulator genes governing bCSC-fate

To define the genetic networks that govern bCSC identity, we carried out a genome-wide RNAi screen using a siRNA pool library targeting ~18,000 human genes (4 pooled siRNAs/gene), in the SUM159 breast cancer cell line (BCL) (Fig 1A). siRNA pools were systematically tested as separate triplicates in an ALDEFLUOR-probed bCSC detection assay (Fig 1B). This approach, adapted from a previous work (El Helou *et al*, 2017), allows the concurrent measurement of changes in bCSC proportion upon gene knockdown (KD). Data were analyzed by calculating the averaged total cell amount and the averaged bCSC proportion upon gene KD, relative to the scrambled siRNA negative control (*Total Cell Amount* and *% bCSC* parameters, Fig 1C–E, Dataset EV1). Following data correction, B-scores of the *%bCSC* parameter were calculated for each targeted gene and were plotted against the normalized bCSC proportion (Fig 1F). A gene was selected as a candidate when its silencing presented an absolute B-Score above or equal to 2.58 (eq. to a *P*-value < 0.01) and an absolute fold change (FC) of bCSC proportion above or equal to 2 in reference to the negative control. siRNA pools designed to target ubiquitous cell survival genes, and that were found to induce a massive cell death (> 80%) of the whole cell population, were excluded from the analysis to focus on hit genes that specifically alter the bCSC-fate (Fig 1C–E). We thus identified 332 hit genes whose inhibition significantly decreased the bCSC proportion (B-score > 2.58; FC ≤ 2; Hit list 1, Dataset EV1). ALDH1A1, which is responsible of the ALDEFLUOR substrate metabolism, was confirmed as a top-ranked hit in this gene list. Conversely, the inhibition of 421 genes resulted in the bCSC expansion (B-score > 2.58; FC > 2; Hit list 2, Dataset EV1) (Fig 1F). This list constitutes a functional core of genes that may regulate bCSC-fate.

## Functional mapping of the bCSC-related processes

To obtain a more comprehensive view of the processes represented by the hit list, we recently developed a program for analysis of biological networks and pathways called High-Throughput Screening-Network (HTS-Net) (Rioualen *et al*, 2017). HTS-Net integrates transcription factors–target genes interaction data (regulome) and protein–protein interaction networks (interactome) on top of screening scoring methods. To detect subnetworks of genes whose inhibition impacted the proportion of bCSCs, we did a HTS-Net analysis with the minimal network score improvement threshold of 0.01 (Fig 2A and B). This network score is the minimum threshold to obtain the maximum of statistically significant subnetworks/genes (*P* < 0.01, Fig 2A and B). This interactome/regulome analysis revealed eleven "self-renewal" subnetworks enriched in regulators of bCSC-fate (Dataset EV2, Fig 2C). We could distinguish five subnetworks enriched in genes whose inhibition induced an expansion of the bCSC population. Among them, we identified components of the KPNA2 regulome, TNF signaling, proteasome, spliceosome, and chromatin remodeling (EP300-related network) complexes. The six remaining subnetworks were enriched with genes whose inhibition induced a depletion of the bCSC population, therefore constituting a potential reservoir for new bCSC therapeutic targets. These subnetworks were associated with nuclear receptors NCOR1/RARA/NR3C1 complex, mediator complex, autophagy, ribosomal machinery, and ADARB1 regulome. All these subnetworks represent a resource to identify new therapeutic targets against the bCSC population.

## Identification of targeted therapeutic compounds with anti-bCSC activity

To identify potentially actionable targets, we selected from the identified subnetworks genes encoding proteins for which chemical inhibitors were listed in public databases (Google search and DrugSurv; Amelio *et al*, 2014), and commercially available. We identified 15 compounds targeting 11 different genes/pathways from 8 subnetworks previously identified (Fig 3A). We first functionally tested the inhibition effect of these 11 genes/pathways on the bCSC population by using tumorsphere-forming efficiency. Using a focused library of 33 siRNAs designed to inhibit the 11 genes/pathways, we confirmed that a reduction of the ALDH[br] cell population was associated with a decrease in tumorsphere-forming efficiency (*R* = 0.68; *P* = 0.04) (Fig 3B). We next tested if a treatment with the selected compounds could mimic the siRNA effect on the bCSC population. We determined the IC50 of each of these drugs on the cell viability of five different breast cancer cell lines (Br-Ca-MZ01, MDA-MB-436, S68, SUM149, and SUM159, Appendix Fig S1). Then, we tested the drugs

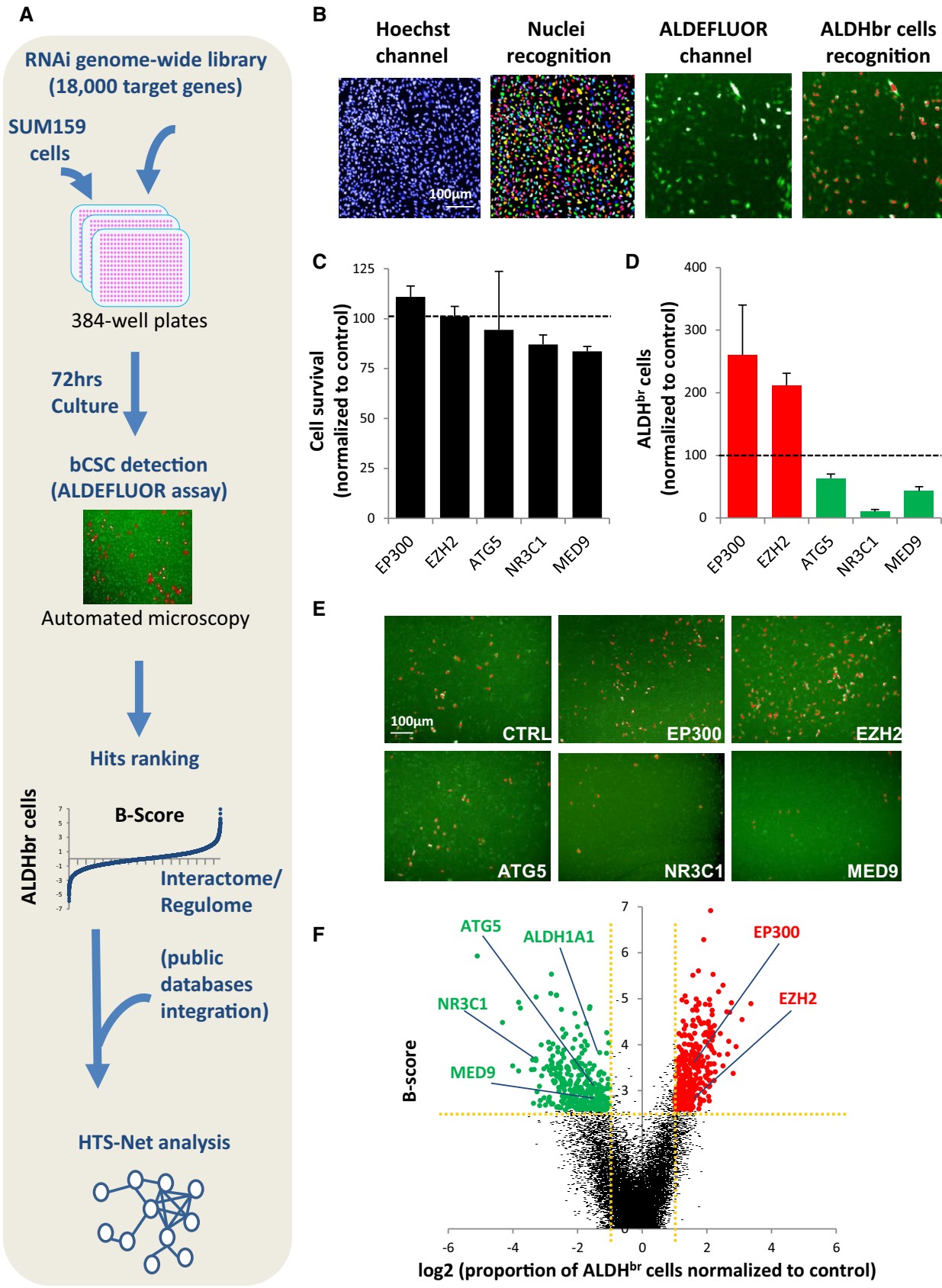

**Figure 1.**

◄

Figure 1.  A genome-wide RNAi screen for the identification of bCSC-fate regulator.

A        Schematic representation of RNAi screening strategy.
B        Detection of ALDH[br] cells by high-content screening. The algorithm first detects nuclear region of interest (ROI) in the Hoechst channel (rainbow areas). Then, it
         computes the averaged ALDH signal in the ROI to identify ALDHbr cells (red nuclear ROIs) and calculate their proportion.
C, D     Example of candidate genes whose silencing modified the ALDHbr cell proportion without inducing a massive cell death (cell viability > 20%, B-score >|2.58|). Data
         represent mean ± SD ($n$ = 3).
E        Representative images of the high-content screening captures. ALDEFLUOR cellular staining is represented in green. ALDHbr cells are labeled in red.
F        Volcano plot of the hits. Scatter points represent genes, the $x$-axis is the log2 fold change of the proportion of ALDHbr cells compared to the control, and the $y$-axis
         is the B-score. Genes targeted by a siRNA inducing a significant reduction (green dots; B-score > 2.58, log2 FC < 1) or increase (red dots; B-score > 2.58, log2 FC > 1)
         of the ALDHbr cell proportion are highlighted.

treatment effect on the bCSC population using a range of concentrations centered on the IC50 to limit unspecific cytotoxic effect (Appendix Fig S2). To determine the drug treatment that could interfere with bCSC-fate in a dose-dependent manner, we developed a composite score (K) calculated from a linear regression model (see Materials and Methods section). Drugs with a positive K-score induced an increase of the bCSC population whereas a negative K-score characterized an anti-bCSC compound (Fig 3C and D). We observed a significant correlation between results mediated by gene silencing and the corresponding treatment with a targeted inhibitor compound ($R$ = 0.57; $P$ = 0.03, Fig 3E). These results support the use of this drug panel as potential inhibitors of the bCSC-fate. To identify the more efficient anti-bCSC compounds, among molecular diversity of breast cancer, we compared drug-response profiles similarities in five different BCLs (Fig 3F). We found that independent drugs with the same annotated molecular target had a highly correlated profile suggesting that the treatment effects occur primarily through inhibition of the intended molecular target. We identified two groups of drugs with a strong reproducible effect among the different tested cellular models. The first group induced a bCSC expansion and corresponds to three epidrugs (I-CBP112 [EP300], tazemetostat [EZH2], and GSK343 [EZH2]). Supporting our results, recent reports have questioned the use of EZH2 inhibitors as an efficient cancer therapy. Indeed, some data suggest that EZH2 inhibition may be detrimental to cancer care by causing a robust switch in cell fate through activation of pluripotency networks, which ultimately promote tumor relapse (de Vries et al, 2015; Wassef & Margueron, 2017). The second group of compounds encompassed three drugs (mifepristone [NR3C1], salinomycin [autophagy], and JQ1 [Mediator complex]) that showed a reproducible anti-bCSC effect in all the BCLs tested. Accordingly, those compounds significantly reduced primary and secondary tumorsphere-forming efficiency, whereas the EZH2-targeted therapy (GSK343) had the exact opposite effect (Fig 3G and H). Based on these results, mifepristone, salinomycin, and JQ1 represent promising anti-bCSC agents with diverse mechanisms of action.

## Salinomycin synergizes with JQ1 to reduce the bCSC population

bCSC self-renewal is the result of a complex combination of intertwined molecular pathways. To tackle this complex rewiring of pathway crosstalk, we proposed to test combination treatment targeting multiple bCSC-related processes. Here, we selected the three compounds found with the most reproducible effect when used as a single anti-bCSC agent. We computed the individual dose–

response matrixes testing pairwise combinations of eight doses of the selected drugs on the four BCLs with the strongest response to single agent (Figs 4A and B, and EV1A and B). In this assay, the fraction affected by drug combinations (FA) was based on the variation of the bCSC/non-bCSC ratio. We calculated synergy distribution of each combination following Loewe mathematical model and using Combenefit tool (Di Veroli et al, 2016) (Figs 4A and B, and EV1A and B). Salinomycin/JQ1 combination had the best synergistic interaction with a reproducible effect on the different BCLs tested. Indeed, we detected a dose–response surface, delineating combination effects in concentration space, with a significant synergistic interaction (Neighborhood $Z$-score > 0; $P$ < 0.05) for all the four BCLs. Similar results were obtained by combining siRNA constructs, with the knockdown of ATG5 and MED12 presenting the more efficient association to reduce the bCSC population (Fig EV1C). We next confirmed the JQ1/salinomycin combination as the more potent anti-bCSC treatment by using tumorsphere assay, with a 2.8-fold reduction of primary tumorsphere-forming efficiency in cells treated with JQ1/salinomycin compared to the control ($P$ < 0.001) and a 5.4-fold reduction of secondary tumorsphere-forming efficiency ($P$ < 0.0001) (Fig 4C and D). Once again, similar results were observed by combining siRNA constructs, with a 4-fold reduction of tumorsphere-forming efficiency in cells simultaneously invalidated for ATG5 and MED12 compared to ATG5 or MED12 alone (Fig EV1D). Altogether, these results support that a combination of JQ1 and salinomycin is a promising therapeutic approach to reduce the bCSC proportion. To further define the potential interest of this drug combination for the treatment of patients with breast cancer, we compared the effects of the cytotoxic agent docetaxel (commonly used for the treatment of breast cancers) to salinomycin and JQ1 treatments. As previously described (Ginestier et al, 2010), docetaxel treatment induced an increase of the bCSC pool in both BCLs tested, as determined by the proportion of ALDHbr cells and the tumorsphere-forming efficiency (Fig EV2). Notably, salinomycin/JQ1 combination prevents the docetaxel-induced bCSC increase in both BCLs tested.

## Salinomycin/JQ1 drug combination reduces the bCSC pool by inducing cell differentiation

The reduction of the bCSC proportion induced by the salinomycin/JQ1 combination may result from different and non-exclusive cell-fate scenarios, including preferential or selective bCSC apoptosis, modification of proliferation rate of one or both cell subpopulations (ALDHbr and ALDH⁻), and increased bCSC differentiation. To

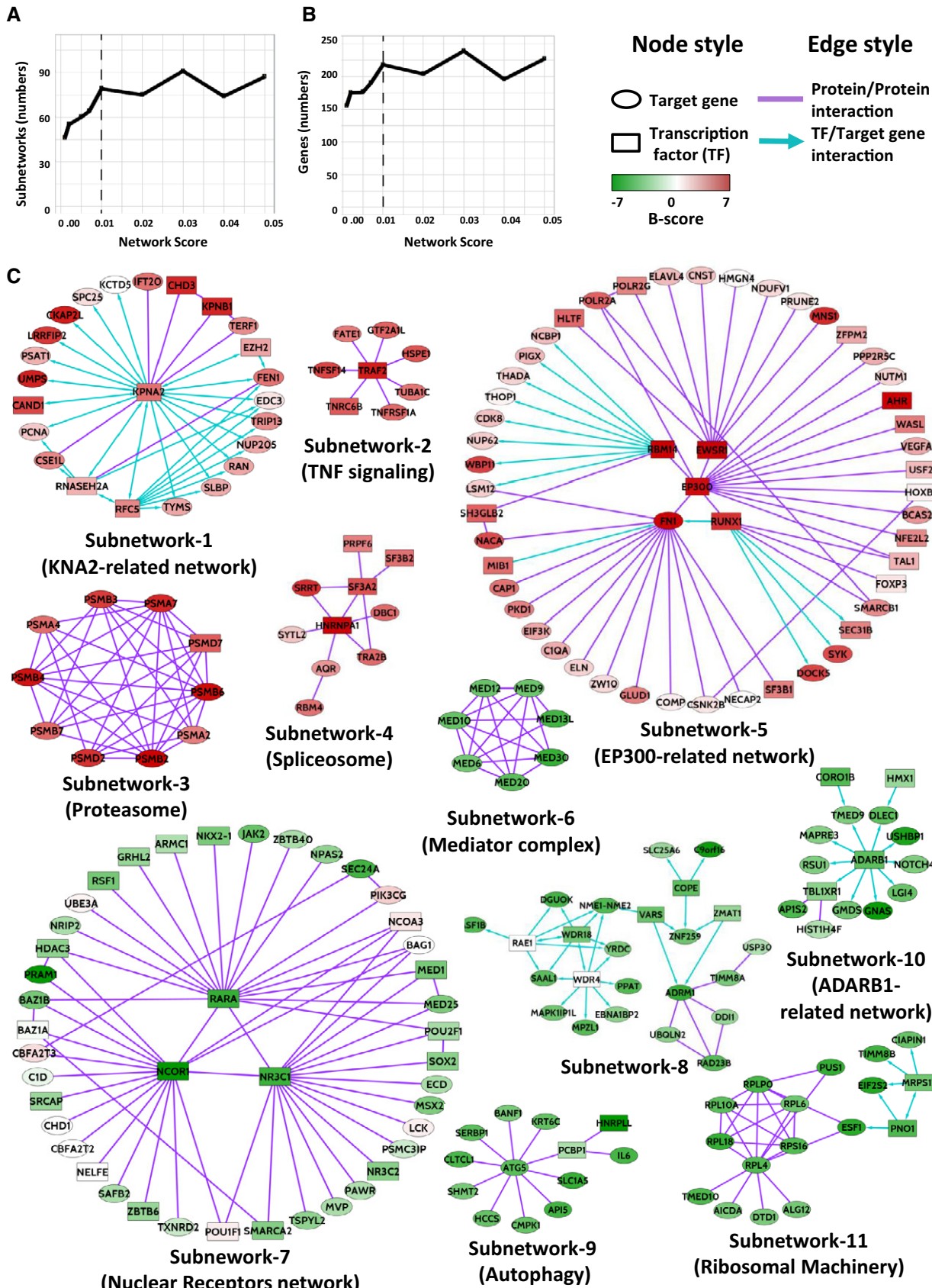

**Figure 2.**

**Figure 2. Interactome/regulome analysis.**

A, B   Selection of the minimal network score improvement threshold to maximize the number of subnetworks containing statistically significant genes. All network scores superior to 0.01 did not present any improvement in the numbers of statistically significant subnetworks (A) or genes (B).

C       The eleven subnetworks with a statistically significant network score (P < 0.01). Five subnetworks are enriched in negative regulators of bCSC-fate (red genes) and six in positive regulators (green genes).

evaluate how drug treatments affect bCSC proportion, we first measured the proportion of apoptotic cells in each of the cell subpopulations after 48 h of treatment. The genotoxic agent cis-diaminedichloroplatine (II) (Cisplatin) was used as positive control and demonstrated a dose-dependent capacity to induce apoptosis in both cell subpopulations (Fig 4E). However, we did not observed any significant increase of apoptotic cells in each cell subpopulations following JQ1 or salinomycin treatment alone or in combination. This result excludes a differential apoptotic rate in any of the cell subpopulations to explain the reduction of the bCSC proportion in response to the treatment. To further explore the cell mechanism underlying the drug-induced bCSC reduction, we developed a lineage tracing system using an engineered SUM159 cell line (Fig 4F). Using this system, we can follow the non-bCSC (ALDH$^-$/ RFP$^+$ cells) and bCSC (ALDHbr/BFP$^+$) progenies under drug treatment (Fig EV3). JQ1, SAL, or combination of both did not induce any modifications of the RFP$^+$/BFP$^+$ cell ratio (Fig 4G), indicating that the drug effect on the bCSC proportion cannot be explained by a bCSC/non-CSC differential proliferation rate. However, we did observe a significant dose-dependent reduction of the ALDHbr cell proportion in the bCSC progenies (BFP$^+$ cells) under treatments compared to the control (Fig 4H and I). A concentration of 50 nM of JQ1 or salinomycin alone was able to induce a two-fold decrease of the ALDHbr/BFP$^+$ cells proportion compared to the control ($P = 0.03$) and a four-fold decrease for the drug combination (Sal [50 nM]/JQ1 [50 nM]); $P = 0.01$) (Fig 4H). As expected, we did not detect any ALDHbr cells in the RFP$^+$ progenies (Fig 4I). Altogether, these results suggest that JQ1 and salinomycin treatments mainly reduce the bCSC proportion by promoting bCSC differentiation in non-bCSC.

## Salinomycin/JQ1 drug combination reduces the bCSC pool in patient-derived xenografts (PDXs)

To further evaluate the impact of salinomycin/JQ1 drug combination on the bCSC population, we utilized three independent patient-derived xenograft models derived from triple-negative breast cancers (CRCM404, CRCM434, and CRCM494). Cells from these PDXs were transplanted orthotopically into fat pads of immunodeficient (NSG) mice. Using these models, we previously demonstrated that the bCSCs were contained in the ALDH$^{br}$ cell population (Charafe-Jauffret et al, 2013). We injected single cancer cell suspension into fat pads of NSG mice and monitored tumor growth. When the tumor size was approximately 80 mm³, we started treatment with salinomycin alone, JQ1 alone, or the combination of both. Tumor growth was compared with that of placebo-treated controls (Fig 5A). Salinomycin had no effect on PDXs growth, at least in the timeframe of our experiments (Figs 5B and EV4). JQ1 or the drug combination treatment tended to reduce tumor growth. This observation is associated with an increase of apoptotic cells in JQ1- and Sal/JQ1-treated tumors without any impact on cell proliferation (Fig EV5A and B). These observations are

concordant with previous reports on anti-bCSC therapy that selectively target bCSCs while mainly sparing actively dividing differentiated cancer cells, ending up with limited short-term effect on tumor growth (Ginestier et al, 2010; Salvador et al, 2013; Tosoni et al, 2017). To detect a potential impact on the tumorigenic cell populations, we first measured the proportion of ALDH$^{br}$ cancer cells after 2 weeks of treatment. PDXs treated with salinomycin or JQ1 alone presented a moderate reduction of the ALDH$^{br}$ cell population. However, we observed a pronounced decrease (2- to 4-fold, $P < 0.01$) of the ALDH$^{br}$ cell population in the tumors treated with the Sal/JQ1 combination (Fig 5C–E). To functionally prove the reduction of the bCSCs population in the treated tumors, we performed a limiting dilution transplantation assay (Fig EV4) in secondary mice. Tumorigenicity is directly related to the presence of CSCs, and this assay gives an estimate of the proportion of residual tumorigenic CSCs (Kreso et al, 2014). For both PDX models, residual cells isolated from treated tumors had a markedly reduced tumor-initiating capacity in secondary mice compared to control (Figs 5F and EV4). As determined by limiting dilution analysis (LDA), bCSC frequency was lower in the salinomycin-treated (bCSC freq.: 1:234; confidence interval [CI]: 69–788 (CRCM404, CRCM434); 1:562, [CI]: 142–2,228 (CRCM494)) or JQ1-treated tumors (bCSC freq.: 1:234, [CI]: 69–788 (CRCM404); 1:562, [CI]: 142–2,228 (CRCM434, CRCM494)) compared to the placebo-treated tumors (bCSC freq.: 1:72; [CI]: 19–280 (CRCM494); 1:37, [CI]: 14–102 (CRCM404), 1:17, [CI]: 5–60 (CRCM434)) (Figs 5G–I and EV4). The drug combination presented the most drastic effect with a frequency of only 1 bCSC out of 2189 residual tumor cells ([CI]: 630–7,606) for CRCM404 or CRCM434 ($P = 3.89e-5$; $P = 1.11e-6$) and 1 bCSC out of 5,475 residual tumor cells for CRCM494 ([CI]: 1,357–22,086, $P = 5.76e-6$). These results, showing an almost complete eradication of bCSC cells in secondary tumors, strongly suggest that the drug combination salinomycin/JQ1 could be a promising anti-bCSC therapy. To further evaluate the potential of the drug combination salinomycin/JQ1 to limit tumor progression, we determined whether the treatment could impact metastasis formation. We infected two PDX models (CRCM404 and CRCM434) with a luciferase lentivirus reporter system and introduced the cells into NSG mice by tail vein injections. 24 hours after cell injection, we started the drug treatments with either salinomycin or JQ1 alone, or the combination of both. As shown in Fig 5J–L, the salinomycin/JQ1 combination significantly reduced metastasis formation in both PDX models (CRCM404, $P = 0.04$; CRCM434, $P = 0.02$) (Fig EV4J). Taken together, these results suggest that the salinomycin/JQ1 drug combination is a potent therapeutic approach for patients with breast cancers.

## bCSC-related subnetworks are associated with prognosis in breast cancer patients

A consequence of these findings may be a direct association between the different bCSC-related subnetworks and tumor progression. We first validated that genes contained in subnetwork 6

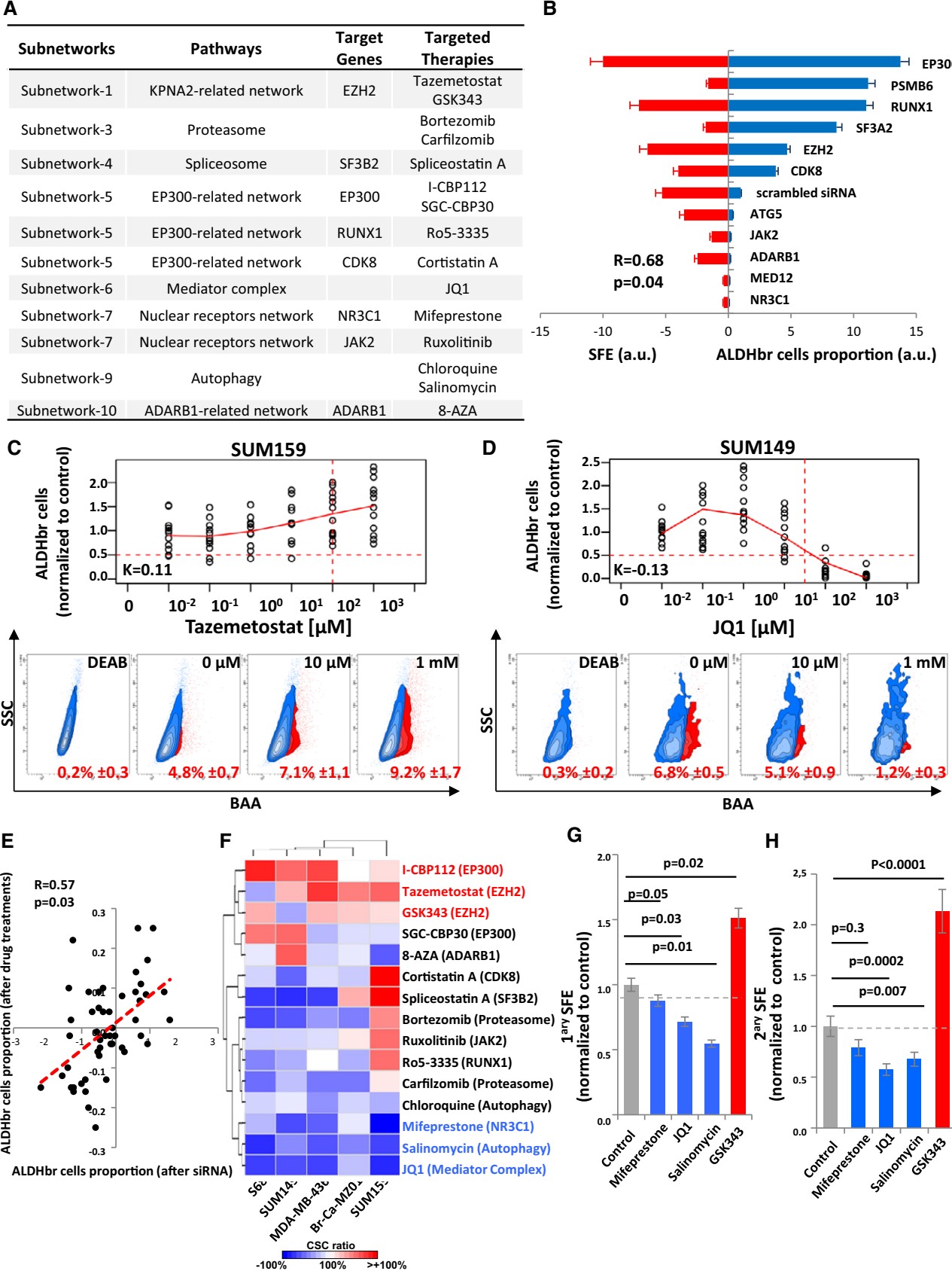

Figure 3.

◄

**Figure 3.  Identification and validation of compounds that modulate bCSC-fate.**

A    Table summarizing the different target genes/targeted therapies identified for each subnetworks.

B    Comparison of bCSC variation after gene knockdown by ALDEFLUOR phenotyping (right) and tumor SFE (left) using the 11 siRNA pools (3 siRNAs/pool) against the selected target genes and one scrambled siRNA pool used as control. Results for each siRNA pool (from top to bottom) are represented as opposite bars (n = 3). Correlations are measured using Spearman rank correlation (R).

C, D   Examples of dose-dependent ALDHbr cells variation in response to drug treatments. ALDHbr cell proportion in response to tazemetostat in SUM159 (n = 10) (C) and JQ1 in SUM149 (n = 10) (D). The horizontal red dashed line corresponds to a reduction of 50% of the ALDHbr cell proportion in the treated conditions compared to the control, and the vertical red dashed line corresponds to the target dose predicted by the K-score. On the bottom panel, corresponding flow charts for the ALDEFLUOR staining. DEAB is an ALDH inhibitor used as negative control.

E    Pearson correlation analysis of the variation of the ALDHbr cells proportion upon drug treatment and gene silencing.

F    Heat map for the variation of the ALDHbr cells proportion under drug treatment, in five different cell lines. Red and blue indicate the K-score in treated conditions, respectively, above and below the K-score in the untreated condition.

G, H   Primary (n = 4) (G) and secondary (n = 4) (H) tumorsphere-forming efficiency (SFE) of SUM159 cells under treated and untreated conditions. Gray dashed line corresponds to the level of SFE in the untreated conditions. Statistical test used is pairwise chi-square test. Data represent mean ± SD.

(Mediator complex, targeted by JQ1) and subnetwork 9 (Autophagy, targeted by salinomycin) presented a significant downregulation in the salinomycin/JQ1-treated tumors compared to the control. We performed metagene analysis for the global expression of genes composing each of the 11 subnetworks identified by the RNAi screen. Using these metagenes, we classified treated tumors and observed a significant and specific low expression level of metagenes generated from subnetworks 6 and 9 in the salinomycin/JQ1-treated tumors compared to the controls (Figs 6A and EV5C). These results support the functional role of these subnetworks in the regulation of the bCSC proportion during tumor progression. Thus, we hypothesized that patient tumors harboring a high expression level of genes that compose subnetworks enriched in positive regulators of bCSC self-renewal (green subnetworks; see Fig 2D), or in negative regulators of bCSC self-renewal (red subnetworks; see Fig 2D) may be associated with a poor or good prognosis, respectively. To test this hypothesis, we focused on subnetwork 5 (EP300-related network), subnetwork 6 (Mediator complex, targeted by JQ1), and subnetwork 9 (Autophagy, targeted by salinomycin). We demonstrated that these subnetworks contained bCSC regulators in different BCLs/PDXs, with valid targets. Using the metagenes for these subnetworks, we classified 36 clinically annotated gene expression data sets corresponding to 2,578 patients with available follow-up (Dataset EV3). Based on the metagene generated from subnetwork 5, we obtained two groups of patients. Patients with a high expression level of this metagene were associated with a better relapse-free survival (RFS) than patients with a low expression level (5-year RFS 67% versus 74%, HR = 0.77, $P = 1.7e-4$) (Fig 6B and C). Conversely, patients with a high expression level of metagene generated from subnetwork 6 and 9 correlated with high histological grade (Grd > 2), large tumor (pT > 2), axillary lymph node metastasis, and triple-negative phenotype (TNBC) and were associated with a worse relapse-free survival compared to patients with low expression level (subnetwork 6, 5-year RFS 68% versus 72%, HR = 1.22, $P = 0.04$; subnetwork 9, 5-year RFS 67% versus 73%, HR = 1.35, $P = 8.6e-4$). Interestingly, if we combined both metagenes, from subnetworks 6 and 9 (for which we identified therapeutic targets with synergistic drug combination), we could stratify patients according to the gene expression level of both molecular processes. Patients with tumors concurrently harboring a higher metagene expression from subnetworks 6 and 9 presented a poorer clinical outcome than patients with low expressing tumors (subnetworks-6/9, 5-year RFS 64% versus 74%, HR = 1.78, $P = 2.4e-4$). A multivariate analysis of expression of subnetworks 6 and 9 based on the

Akaike information criterion (AIC) highlighted cooperation between these two gene networks in prognostic term ($P = 9.88e-4$). Altogether, these observations suggest that these bCSC-related subnetworks represent clinically relevant players in breast cancer progression.

# Discussion

The success of anticancer therapy is usually limited, due to the presence of drug-tolerant CSCs. Indeed, accumulating evidence has shown that CSCs are more resistant than non-CSCs to various types of conventional therapies (Shibue & Weinberg, 2017). Understanding the intrinsic molecular mechanisms that define this drug-tolerant state holds the promise of yielding novel therapeutic strategies to extend clinical remissions and prevent recurrence. To achieve this goal, we have developed a high-throughput functional assay to perform a human genome-wide gene loss-of-function screen, with bCSC-fate as a readout. Using this strategy, we identified key genes governing bCSC-fate and did a functional mapping of the associated biological processes. We noted several functional units directly linked to the transcriptional regulation, with upstream effectors such as EP300-related network (subnetwork 5) or mediator complex (subnetwork 6), and post-transcriptional effectors with the ADAR-related network (subnetwork 10) or the spliceosome machinery (subnetwork 4). These processes could directly participate to the transcriptional regulatory circuit governing the bCSC self-renewal program. Some of them have been recently linked to the regulation of normal and malignant stem cell fate. RNA editing mediated by ADAR editases (A-to-I editing) has been shown to play a key role in hematopoietic and intestinal stem cell maintenance (Zipeto et al, 2015). Moreover, ADAR1 activation drives leukemic stem cell (LSC) self-renewal and its inhibition appeared as a good strategy to impair blast expansion in chronic myeloid leukemia (Jiang et al, 2013; Zipeto et al, 2016). These results are concordant with our observations suggesting a decrease of the bCSC population following ADARB1 inhibition. The mediator complex also appears as a pivotal regulator of the self-renewal program through activation of normal and malignant hematopoietic stem-cell-specific super-enhancers (Pelish et al, 2015; Aranda-Org et al, 2016). Interestingly, the mediator subunits MED1 and MED24 are essential to pubertal mammary gland development suggesting a role in the mammary stem cell fate (Hasegawa et al, 2012). Accordingly, we observed a drastic reduction of the bCSC population following mediator complex inhibition.

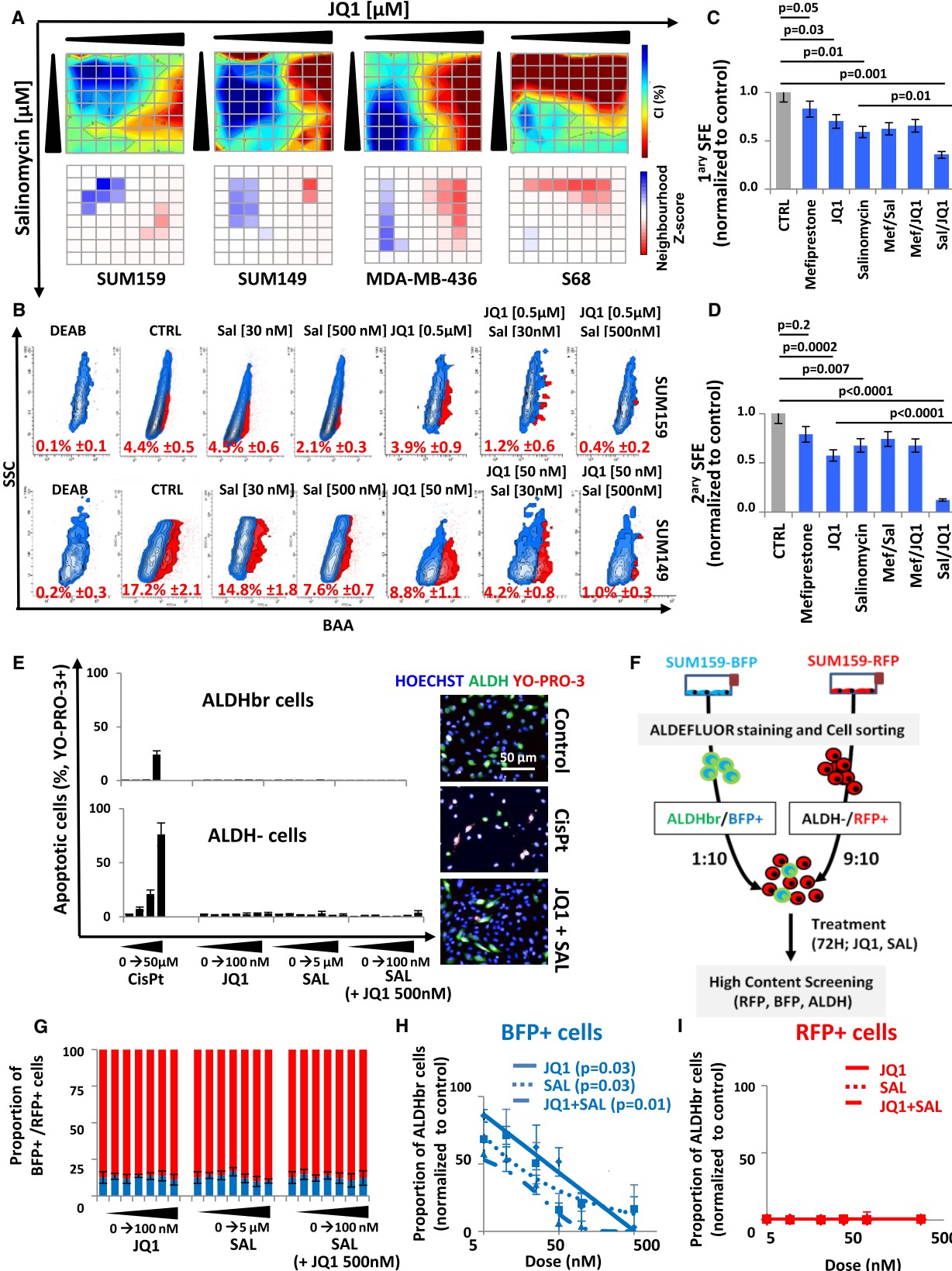

Figure 4.

**Figure 4. Drug combination screen identifies salinomycin as acting synergistically with JQ1 to reduce the bCSC proportion.**

A   Interaction surface built using the estimated combination index (CI) for the salinomycin and JQ1 drug combination, in four different breast cancer cell lines (top panels). The CI is represented in 2D color code interaction surface. The color code spans from strong synergism (dark blue, CI < 1) to strong antagonism (red, CI > 1). The concentrations of each particular drug in the combinations are denoted on each axis. On the bottom panels, neighborhood $Z$-core matrix estimating the significant synergistic interaction. The color code spans from a statistically significant synergism (dark blue, CI < 1, $P$ < 0.01) to a statistically significant antagonism (dark red, CI > 1, $P$ < 0.01).

B   Representative examples of flow chart for the ALDEFLUOR staining following different JQ1/salinomycin drug combinations. DEAB is an ALDH inhibitor used as negative control. Data represent mean $\pm$ SD.

C, D   Primary ($n$ = 4) (C) and secondary ($n$ = 4) (D) tumorsphere-forming efficiency (SFE) of SUM159 cells under mono- and combined therapy compared to untreated conditions. Statistical test used is pairwise chi-square test. Data represent mean $\pm$ SD.

E   Bar plots representing the dose-dependent changes in the proportion of apoptotic cells (YO-PRO-3$^+$) in each cell subpopulations, in response to drug treatments (right panel). Cisplatin (CisPt) is used as a positive control. On the right panel, representative fluorescence images obtained in different treatment conditions ($n$ = 6). Nucleus are in blue (Hoechst), bCSCs are in green (ALDH), and apoptotic cells are in red (YO-PRO-3). Data represent mean $\pm$ SD.

F   Schematic representation of the lineage tracing system protocol.

G   Representation of the BFP$^+$/RFP$^+$ cells proportions for increasing drug concentrations of either JQ1 or salinomycin alone, or in combination. Blue bars: BFP$^+$ cells proportion; red bars: RFP$^+$ cells proportion ($n$ = 3). Data represent mean $\pm$ SD.

H, I   Representation of the bCSC proportion in the BFP$^+$ (H) and RFP$^+$ (I) progenies in the control cells compared to the JQ1- and salinomycin-treated cells alone or in combination ($n$ = 4). Statistical test used is Student's $t$-test. JQ1 versus CTRL and Sal versus JQ1 $P$ = 0.03; JQ1/Sal versus CTRL $P$ = 0.01. Data represent mean $\pm$ SD.

Source data are available online for this figure.

All these recent reports support the validity of our interactome/regulome mapping analysis, which provides an important resource for further functional identification of key drivers of bCSC-fate.

Our functional mapping is also a source of potential actionable targets for the development of anti-bCSC therapies. In this study, we have focused on a panel of commercially available biologically active compounds, from which we identified mifepristone (NR3C1 inhibitor), salinomycin (autophagy modulator), and JQ1 (mediator complex disruptor) as the most potent bCSC inhibitors. Salinomycin, an ionophoric natural product, identified as one of the first anti-bCSC compound (Gupta et al, 2009), through inhibition of the autophagic flux (Yue et al, 2013), can be considered as a positive control, thus validating our approach. Recently, we deciphered the mechanism of action of salinomycin on the bCSC population. Indeed, salinomycin changes iron cellular homeostasis by promoting sequestration of iron in lysosomes and ultimately perturbates autophagic flux (Mai et al, 2017). Mifepristone, an anti-glucocorticoid, and JQ1, a BET bromodomain inhibitor, however may represent new promising anti-bCSC therapies. Different mechanisms of action may explain the efficiency of these compounds on the bCSC population. The transcriptional circuitries that govern stem cell identity are finely tuned by dynamic chromatin remodeling. The large open-chromatin domains, so-called "super-enhancers" (SEs), have a central role in the control of the expression of particularly important genes in stem cell behavior (Adam & Fuchs, 2016). These SEs are densely loaded with mediator complex driving expression of stem cell-related genes (Pelish et al, 2015). JQ1 treatment causes a rapid release of mediator from SEs inducing a transcriptional repression of the genes implicated in stem cell identity (Bhagwat et al, 2016; Yokoyama et al, 2016). Similar results were obtained with the silencing of mediator subunits suggesting that our observations in bCSC may be mediated through an inhibition of SE-regulated genes. Moreover, this mechanism of action is in accordance with the induction of bCSC differentiation suggested by our study. Interestingly, JQ1 has been identified as a potent therapy for the treatment of triple-negative breast cancers, which are known to be enriched in bCSCs (Shu et al, 2016).

Beside activation of intrinsic molecular pathways, bCSC-fate is also highly regulated by local microenvironments. External signals from the tumoral niche, such as hormones and cytokines, have been found to drastically regulate bCSC behavior (Batlle & Clevers, 2017). Among them, glucocorticoids have been identified as activators of the hippo/YAP pathway, a major regulator of the self-renewal program (Zanconato et al, 2016; Sorrentino et al, 2017). These observations may suggest that mifepristone could mediate its effect on the bCSC population through YAP inhibition. Interestingly, mifepristone has been successfully used in the therapeutic arsenal of chemotherapy-resistant triple-negative breast cancers (TNBCs) (Skor et al, 2013). Further studies are needed to firmly identify the mechanism of action of these anti-bCSC compounds on the bCSC-fate.

**Figure 5. JQ1/Salinomycin combination decreases the bCSC population in patient-derived xenografts.**

A   Schematic representation of the in vivo experimental design.

B   Effect of JQ1 and salinomycin treatment on the tumor growth of CRCM434 ($n$ = 10) (see Fig EV4A and E for CRCM404 and CRCM494). The gray area corresponds to the period of drug treatments. Data represent mean $\pm$ SD.

C–E   Quantification of the proportion of residual ALDHbr cells in treated PDX normalized to the proportion of ALDHbr cells in the vehicle-treated tumors, measured by flow cytometry. Statistical test used is Student's $t$-test. Data represent mean $\pm$ SD ($n$ = 10).

F–I   Reimplantation assay (CRCM434). Two-week treated PDXs were reimplanted, in serial dilutions, into new recipient mice, and tumor growth was monitored (F, see Fig EV4 for CRCM404 and CRCM494). Each curve represents the growth kinetic from one individual injection. (G–I) Bar plots display bCSC frequency calculated using an extreme limiting dilution analysis (ELDA). Results are expressed as the estimated number of bCSCs for 10,000 tumor cells. Statistical test used is pairwise chi-square test.

J–L   Effect of JQ1 and salinomycin treatments on the metastasis formation of CRCM404 and CRCM434. Metastasis formation was monitored using bioluminescence imaging in the treated mice livers and lungs (J). Quantification of the normalized photon flux revealed a statistically significant decrease in metastasis formation in Sal/JQ1-treated mice compared to the control mice (K, CRCM434, $P$ = 0.04; L, CRCM404, $P$ = 0.02). Statistical test used is a Wilcoxon test. Data represent mean $\pm$ SD ($n$ = 10).

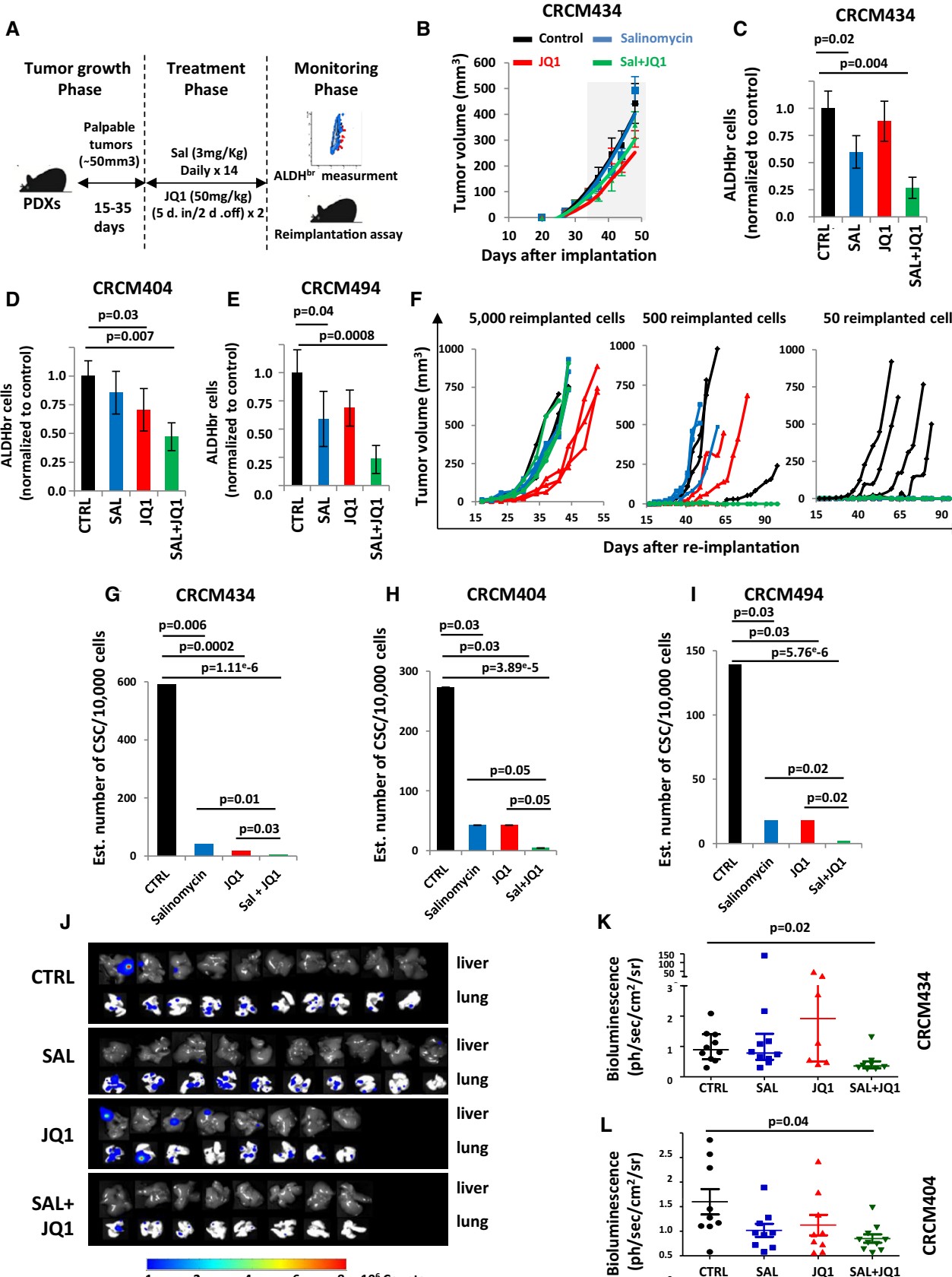

Figure 5.

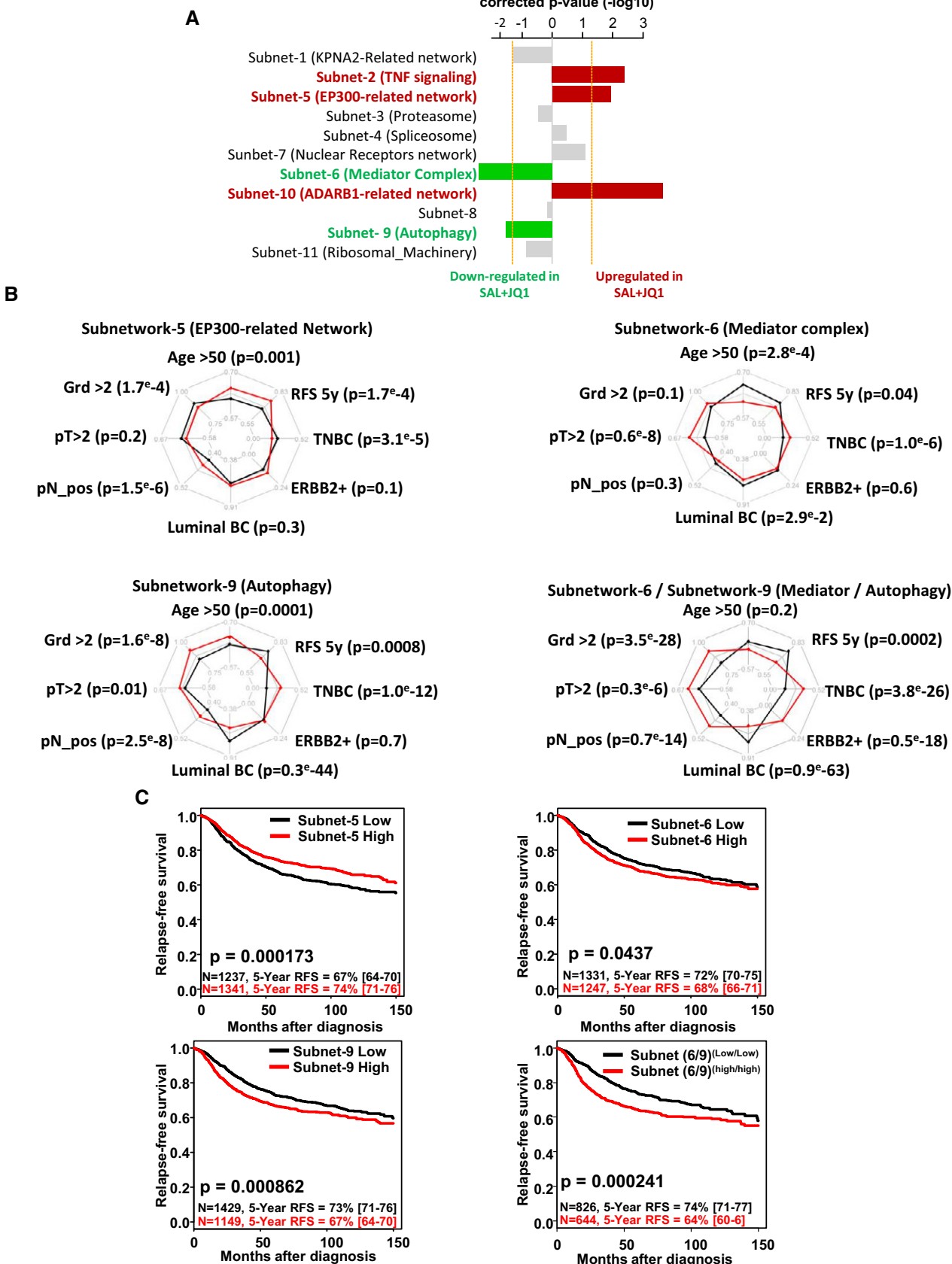

Figure 6.

**Figure 6. Clinical association between subnetwork metagene expression and breast cancer prognosis.**

A  Bar plot representing the corrected *P*-value of the logistic regression comparing the gene expression level of each subnetworks in salinomycin/JQ1-treated tumors compared to the control. Subnetworks presenting a significant downregulation (green bars, *P* <−1.3) or upregulation (red bars; *P* > 1.3) of their metagenes in the salinomycin/JQ1-treated tumors are highlighted.

B  Radar charts comparing clinicopathological features for the patients classified according to the expression of the metagene from the subnetworks 5, 6, and 9 and the combination of subnetworks 6 and 9. Each axis of the diagrams represents a scale of proportions for a specific feature, ranging from 0 to 1. *, chi-square test, *P* < 0.0001.

C  Kaplan–Meier relapse-free survival curve according to subnetwork metagenes expression levels in breast tumors. Curves were compared with a log-rank test (*P*-value).

While the self-renewal machinery provides the basis for a new therapeutic approach in the treatment of breast cancer, 15 years of research on bCSCs have also revealed that stemness results from the convergence of cell-intrinsic features and local signals. Our observations are concordant with these findings, with the identification of eleven independent functional subnetworks driving bCSC-fate. In this context, the future development of successful clinical strategies will be tightly linked to optimal drug combinations to tackle these intertwined pathways. In this study, we validated in pre-clinical models the salinomycin/JQ1 drug combination as an optimal association to reduce the bCSC population. Interestingly, similar observations have been done in acute myeloid leukemia, with inhibition of autophagy that sensitizes leukemic stem cells to JQ1 (Jang *et al*, 2017). These results suggest the potential clinical utility of this specific drug combination to target bCSC in triple-negative breast cancers and possibly CSCs from other types of cancer.

# Materials and Methods

## Ethics statement

Samples of human origin and the associated data were obtained from the IPC/CRCM Tumour Bank that operates under authorization # AC-2013-1905 granted by the French Ministry of Research ("Ministère de la Recherche et de l'Enseignement Supérieur"). Prior to scientific use of samples and data, patients were appropriately informed and filed a written consent, in compliance with French and European regulations. The experiments were conformed to the principles set out in the WMA Declaration of Helsinki and the Department of Health and Human Services Belmont Report. Animal studies were conducted in agreement with the French Guidelines for animal handling and approved by local ethics committee (Agreement no. #16487-2018082108541206 v3).

## Cell culture

Five human breast cancer cell lines from three distinct molecular subtypes (SUM149-Br-Ca-MZ10/Basal, SUM159-MDAMB436/Mesenchymal, and S68/Luminal) were used in this study. SUM149 and SUM159 were obtained from Dr. S. Ethier's (Karmanos Cancer Center, Detroit, MI, USA), S68 from Dr. V. Castros (Université de Rennes, Rennes, France) and BrCa-MZ-01 from Dr V. Möbus (University of Ulm, Germany), and MDA-MB-436 from the American type culture collection (http://www.lgcstandards-atcc.org/). All BCLs were grown in standard medium as previously described (Charafe-Jauffret *et al*, 2009).

## Drugs

BCLs were continuously treated for 72 h in adherent conditions with bortezomib (stock concentration SC = [10 mM], Selleckchem), carfilzomib (SC = [10 mM], Selleckchem), chloroquine (SC = [10 mM], Selleckchem), Cortistatin A (SC = [100 μM], a kind gift from Prof. P. Baran, The Scripps Research Institute, La Jolla, CA, USA), docetaxel (SC = [10 mM], GSK343 (SC = [1 mM], Active Biochemicals Co), I-CBP112 (SC = [10 mM], Sigma), JQ1 (SC = [10 mM], Selleckchem), mifepristone (SC = [10 mM], Selleckchem), Ro5-3335 (SC = [10 mM], Calbiochem), ruxolitinib (SC = [10 mM], Selleckchem), SGC-CBP30 (SC = [10 mM], Selleckchem), salinomycin (SC = [200 μM], Selleckchem), spliceostatin A (SC = [10 mM], Adooq Biosciences), tazemetostat (SC = [5 mM], Selleckchem), and 8-AZA (SC = [100 mM], Chembiotech). All these compounds were resuspended in dimethyl sulfoxide (DMSO; Sigma), except for chloroquine resuspended in $H_2O$. For the *in vivo* experiments, salinomycin (SC = [6 mg/ml], Medchemexpress) and JQ1 (SC = [100 mg/ml], Medchemexpress) were resuspended in a solution of DMSO/(2-Hydroxypropyl)-β-cyclodextrin (HPCD) 10% (1:9, v/v).

## Cell transfection and miniaturized ALDEFLUOR assay

We performed a systematic, individual, and transient gene loss-of-function screening in the SUM159 cell line to identify genes regulating its ALDH[br] subpopulation. To achieve this, we used a human genome-wide siRNA library constituted of pooled siRNAs (4 siRNAs/pool) arrayed in 384-well format and designed to specifically target and knockdown 17,785 human genes (pooled On-Target Plus siRNAs, human genome-wide library, Dharmacon). For screening purpose, an automated reverse transfection protocol was developed on a robotic workstation equipped with a 96-well head probe (Nimbus, Hamilton). Briefly, siRNA pools were lipoplexed with Lipofectamine RNAiMAX (Life Technologies) in collagen-coated, clear bottom, black-walled 384-well culture plates (Greiner μClear plates, Cat# 781091). After 15 min of complexation, SUM159 cells were seeded on top of the lipoplexes (1,000 cells/well; final [siRNA] = 20 nM) and incubated for 3 days at 37°C and 5% $CO_2$ in a humidified incubator. Each pooled siRNA from the library was transfected as a separate triplicate in different well positions of three independent culture plates to minimize positional errors. Each culture plate also received different positive and negative controls: Eight wells received the transfection reagent alone ("MOCK" well, negative controls), sixteen were transfected with a pool of four scrambled siRNAs ("NEG" Wells, negative control, ON-TARGETplus Non-targeting Pool, Dharmacon), and four were transfected with a pool of cytotoxic siRNAs ("AllStars" wells, positive control, Allstars

maximal death control, Qiagen). Additionally, four wells were left untreated to receive the DEAB control during the ALDEFLUOR assay (see below).

Three days post-transfection, SUM159 cell amount and the %ALDH$^{br}$ cell amount (=%bCSC) upon gene knockdown were assessed using a previously described adaptation of ALDEFLUOR assay (Stem Cell technologies) for image acquisition and analysis in microplate format (El Helou et al, 2017). Briefly, after culture supernatant removal, 12 μl of an ALDEFLUOR/Hoechst mixture (13 μl of BAAA substrate per ml of ALDEFLUOR buffer + Hoechst 33342 [Sigma] at 25 μg/ml) was added in the culture wells. As mentioned above, four control wells per plate received N, N-diethylaminobenzaldehyde ("DEAB" wells), an inhibitor of ALDH. After 45 min of incubation at 37°C, the mixture was replaced with 50 μl of ice-cold ALDEFLUOR buffer and plates were immediately imaged on a High Content Imaging device (Operetta HCS epifluorescence microscope, Perkin Elmer). Plates were maintained at ice-cold temperature for the whole duration of the acquisition. Three fields per well were acquired at 10× magnification, in two fluorescence channels: green for ALDEFLUOR (ex: 470 ± 10 nm; em: 525 ± 25 nm) and blue for Hoechst 33342 (ex: 380 ± 20 nm; em: 445 ± 35 nm).

An automated algorithm was developed under Harmony 3.0 (Perkin Elmer) to quantify the *Total cell amount* and the *%bCSC*. Briefly, nuclear regions of interest (ROI), segmented in the Hoechst channel, were used to quantify the *Total cell amount*. Cells were defined as ALDH$^{br}$ when their average background-corrected ALDEFLUOR signal in the nuclear ROI was found above the one measured in the DEAB condition. *%bCSC* was computed as the amount of ALDH$^{br}$ cells over the *Total cell amount*.

### Data preprocessing and hit selection

Data preprocessing was performed in R (R Foundation for Statistical Computing, Vienna, Austria. https://www.R-project.org/). For each culture plate, the *Total cell amount* and the *%bCSC* measured in sample wells were first normalized to the averaged values measured in their respective negative control (NEG) wells. Normalized results were labeled as *Relative Total Cell amount* and *Relative %bCSC*. A time-dependent decay of the ALDEFLUOR signal was found to induce a slight decrease of the *Relative %bCSC* measured over the course of plate acquisitions. To mathematically estimate and correct this decay, we setup a simple non-linear polynomial regression model to fit, plate-by-plate, the relationship between the median *Relative %bCSC* per column and the corresponding column index. For a considered column index, a multiplicative offset was then calculated as the ratio between the median *Relative %bCSC* in the plate and the fitted value at the column index. These multiplicative offsets were then applied column-wise to correct each individual *Relative %bCSC* values. The corrected results were labeled as *Corrected Relative %bCSC*. Analysis of the *Corrected Relative %bCSC* results showed a non-Gaussian, long-tailed distribution of the sample population values. We decided to apply a Box–Cox transformation to this population to achieve normality of the distribution. The optimal λ coefficient for the Box–Cox transformation was determined by fitting a linear regression to quantile-to-quantile (QQ) plots, constructed from quantiles of the Box–Cox transformed *Corrected Relative %bCSC* distribution plotted against quantiles of the corresponding theoretical Gaussian distribution. An optimal

λ = 0.2 was determined to achieve the best linear fit. Normality of the Box–Cox transformed distribution was confirmed by a Kolmogorov–Smirnov test ($P < 0.05$), and transformed results were labeled as *Box–Cox Corrected Relative %bCSC*. *Box–Cox Corrected Relative %bCSC* sample results were then statistically scored using the sample-based B-Score method (Brideau et al, 2003). Briefly, this scoring approach relies first on the iterative application of a two-way median polish algorithm to correct plate values for systematic positional effects. In a second step, the algorithm scores each individual residual well value to indicate its deviation, in terms of median average deviations, from the median of the sample residuals population. Accordingly, absolute B-Scores above 2.56 correspond to values significantly different from the sample population, with an associated $P$-value < 0.01.

### Network building and subnetwork detection

We built, as described previously (Rioualen et al, 2017), a protein–protein interaction network (interactome) by downloading, parsing, and merging the following databases: the Database of Interacting Proteins (DIP), the Human Protein Reference Database (HPRD), the Interologous Interaction Database (I2D), IntAct, the Molecular INTeraction database (MINT), and Human ProteinPedia. A total of 14,423 genes and 171,146 interactions were gathered in the interactome network. We also added TF-TF interaction from TRANSFAC, the Atlas of combinatorial transcriptional regulation, the interactions detected using predicted DNA binding affinity (Mysickova & Vingron, 2012).

We then built a TF-target gene network (regulome), by aggregating the following sources: the Integrated Platform of mammalian Transcription Factors (ITFP), the Transcriptional Regulatory Elements Database (TRED), TRANSFAC, ORegAnno, and The PAZAR database of gene regulatory information. A total of 9,755 genes and 68,703 interactions were mapped in the regulome network.

For each network (interactome and regulome), we used the High-Throughput Screening-Network (HTS-Net) algorithm (Rioualen et al, 2017, http://mobylehome.marseille.inserm.fr/cgi-bin/portal.py#forms::HTS-Net) in order to detect subnetworks of genes whose inhibition significantly impacted the proportion of bCSCs, by associating network nodes with previously computed B-scores. It consists in investigating each node of the considered network as a "seed" for a potential subnetwork, keep it as a candidate if its score is above a fixed threshold, and recursively aggregate neighboring nodes that increase the overall score of the subnetwork by a given threshold *th*. Once the score cannot be improved, the subnetwork is kept aside and other seeds are processed. For the present analysis, we choose a value of *th* = 0.01.

Subnetworks obtained from the interactome analysis and from the regulome analysis are compared against two types of randomized subnetworks in order to be statistically validated. The first type of randomized subnetworks is issued from a shuffled interactome/regulome, and the second type is issued from a shuffled set of B-scores. For each type, 300 randomized subnetworks were generated. The randomized subnetwork score distribution gives a minimal statistically validated score associated with a $P$-value. For both networks, we kept the subnetworks whose score corresponds to a $P$-value of 0.01 or less. The retained subnetworks are then merged

depending on their number of common nodes $n$, giving rise to so-called *meta-subnetworks*. For the present analysis, we choose a network connectivity value of $n = 2$.

## Cell viability and proliferation assays

Inhibitory concentrations 50 (IC50s) were evaluated using 3-(4,5-dimethylthiazol-2-yl)-5-(3-carboxymethoxyphenyl)-2-(4-sulfophenyl)-2H-tetrazolium (MTS) assay (Promega). BCLs were plated in adherent conditions in 96-well plates at 5,000 cells per well, except for SUM159 plated at 3,000 cells per well. After 24 h, treatment with serial dilutions of drugs was started. The effect of treatment on cell viability was estimated after 72 h by addition of 100 µl MTS solution (5 mg/ml in PBS) in each well. Cells were then incubated for 1 h at 37°C. Absorbance was measured at 540 nm in a plate reader (Tecan). Absorbance in treated conditions was normalized with absorbance in control condition to determine concentration–response curves and approximate IC50 concentrations.

## ALDEFLUOR assay

The analysis was processed on single-cell suspension from cell lines or PDXs. The ALDEFLUOR Kit (Stem Cell Technologies) was used to isolate the population with high aldehyde dehydrogenase enzymatic activity using an LSR2 cytometer (Becton Dickinson Biosciences) as previously described (Ginestier *et al*, 2007). To eliminate cells of mouse origin from the PDXs, we used staining with an anti-H2Kd antibody (#553563, BD Biosciences, 1:200, 20 min on ice) followed by staining with a secondary antibody labeled with phycoerythrin (PE; #115-116-146, Jackson Laboratories, 1:250, 20 min on ice).

## Tumorsphere assay

Cells were treated with drugs/siRNAs in adherent conditions for 72 h. For the drug combinations, optimal doses were determined according to the synergies distributions (Mif/Sal, 62 nM/62 nM; Mif/JQ1, 62 nM/0.5 µM; Sal/JQ1, 62 nM/62 nM). Docetaxel was combined with other drugs at its IC50 concentration (i.e 2 nM). Then, treated cells were plated in single-cell suspension in 96-wells ultra-low attachment plates, in a serum-free mammary epithelium basal medium (Ginestier *et al*, 2007). Secondary tumorspheres were generated from subsequent cultures after dissociation of primary tumorspheres. The frequency of cancer cells with tumorsphere-forming potential was determined using the Extreme Limiting Dilution Analysis by plating cells at 25/10/5/3/2 and 1 cell per well ($n = 18$–36 wells/conditions). The number of wells containing at least one sphere after 10 days of culture was considered as positive.

## Apoptotic cell assay

SUM159 cells were seeded into the wells of collagen-coated, black-walled, clear bottom 96-well plates (Costar #3904) at a density of 3,000 cells/well and incubated for 72 h. 6, 24, and 48 h post-seeding, cells were treated with increasing concentrations of JQ1, salinomycin, or combinations of both. Cis-diaminedichloroplatine (II) (Cisplatin, Sigma) and DMSO (Sigma) were included as positive and negative controls, respectively. Control DMSO concentration was set at 0.1% [v/v], corresponding to the highest vehicle

concentration used in the dose responses. 72 h post-seeding, supernatants were removed and wells were treated with 35 µl of a mixture containing ALDEFLUOR (7.5 µl/ml), YO-PRO-3 (Thermo-Fisher Scientific, 1/4,000), and Hoechst 33342 (Sigma, 10 µg/ml) diluted in ALDEFLUOR buffer. When necessary, DEAB was included in the mixture as a control for the ALDEFLUOR assay. After 30 min of incubation at 37°C, the assay reagents were replaced with ice-cold ALDEFLUOR buffer and plates were immediately imaged using a High Content Imaging microscope (Operetta, Perkin Elmer). Plates were maintained at ice-cold temperature for the whole duration of the acquisition. Nine fields per well were acquired at 10× magnification, in three fluorescence channels: green for ALDEFLUOR (ex: $470 \pm 10$ nm; em: $525 \pm 25$ nm), blue for Hoechst 33342 (ex: $380 \pm 20$ nm; em: $445 \pm 35$ nm), and far red for YO-PRO-3 (ex: $630 \pm 10$ nm; em: $705 \pm 55$ nm). The relative %bCSC amount was quantitated as described earlier, with an additional computation step to detect the YO-PRO-3 positive cells. Briefly, YO-PRO-3 signal in the nuclear ROIs was quantitated in the far red channel, and cells were defined as YO-PRO-3 positives when their average YO-PRO-3 signal in the nuclear ROI was found above the one measured in the untreated control. The relative amount of dead/dying cells was computed as the amount of YO-PRO-3 positive cells over the total cell amount.

## Lineage tracing system

SUM159 cells were transduced with commercially available lentiviral particles (Vectalys) to engineer two different cell lines expressing stably and constitutively either Tag-BFP ("SUM159-BFP") or Turbo-RFP ("SUM159-RFP") transgenes under control of EF1a promoter. To extemporaneously create a chimeric cell line, these two cell lines were first labeled with ALDEFLUOR (as described earlier), and FACS-sorted (MoFlo ASTRIOS, Beckman-Coulter). bCSCs (5% brightest cells in ALDEFLUOR channel) and non-bCSCs (10% dimmest cells in ALDEFLUOR channel) were isolated from the SUM159-BFP and SUM159-RFP cell lines, respectively, and mixed together to reconstitute a Chimeric SUM159 cell line by pooling 10% of $BFP^+/ALDH^{br}$ cells to 90% of $RFP^+/ALDH^-$ cells. The chimeric cell line was immediately seeded at a density of 3,000 cells/well into the wells of a collagen-coated, black-walled, clear bottom 96-well plates (Costar #3904) and treated as previously described. 72 h post-seeding, supernatants were removed and cells were labeled and imaged as previously described. Nine fields per well were acquired at 10× magnification, in four fluorescence channels: green for ALDEFLUOR (ex: $470 \pm 10$ nm; em: $525 \pm 25$ nm), blue for Tag-BFP (ex: $380 \pm 20$ nm; em: $445 \pm 35$ nm), red for Turbo-FFP BFP (ex: $535 \pm 15$ nm; em: $595 \pm 35$ nm), and far red for DRAQ5 (ex: $630 \pm 10$ nm; em: $705 \pm 55$ nm). The relative % bCSC amount was quantitated, as described earlier, in the Tag-BFP and Turbo-RFP subpopulations.

## Animal models

To explore the efficiency of treatments on tumor growth, we utilized three primary human breast cancer xenografts (patient-derived xenograft, PDX) generated from three different patients (CRCM404, CRCM434, and CRCM494). These PDXs were generated from chemo-naïve triple-negative breast tumors (Charafe-Jauffret *et al*,

2013). For each PDX, cells from these PDXs were transplanted orthotopically into fat pads of NSG mice without cultivation *in vitro*. We injected 100,000 cells per fat pads of NSG mice (with two injected fat pads per mice) and monitored tumor growth. When tumor size was approximately 80 mm$^3$, we initiated treatment with salinomycin alone (i.p, 3 mg/kg, daily, 14 days), JQ1 alone (i.p., 50 mg/kg, 5 out of 7 days, 2 weeks), salinomycin/JQ1 combination, or placebo injected with a solution of DMSO/HPCD 10% (1:9, v/v). Five mice (i.e., ten tumors) were injected for each PDX and for each group. After 2 weeks of treatment, mice from each group were sacrificed according to ethic statements. Tumors were dissociated, and cells were analyzed for the ALDEFLUOR phenotype. Cells from treated tumors were reimplanted into secondary NSG mice with injection of 50, 500, 5,000 cells for each treated tumor to functionally evaluate the proportion of residual bCSCs in each group of treatment (Ctrl, Sal, JQ1, Sal/JQ1) from the 2 different PDXs. Each mouse that present a tumor reaching a size of 10 mm was considered as a tumor-bearing mouse. To study the efficacy of treatments on metastatic development, we carried out models of experimental metastases by intravenous injections of PDX cells (sorted on H2Kd- and cell viability) into the caudal vein of mice. The day after the cell injection, we started drug treatments as previously described. Briefly, luciferase-expressing CRCM404 and CRCM434 cells (100,000 cells suspended in 100 μl of PBS) were inoculated in the tail vein of NSG mice. After 2 weeks of treatment (10 animals per group), mice were sacrificed and lungs and liver were assayed by bioluminescence. Bioluminescence analysis was performed using PhotonIMAGER (BiospaceLab), following intraperitoneal injection of endotoxin-free luciferin (30 mg/kg).

## Immunostaining on tumor sections

Tumor slides were deparaffinized in xylene and rehydrated in graded alcohol. Antigen enhancement was performed by incubating the sections in citrate buffer pH 6 (Dakocytomation, Copenhagen, Denmark). Staining was done using Peroxidase histostain-Plus Kit (Zymed) according to the manufacturer's protocol. Ki67 antibody (#R626, RTU, Agilent) and cleaved caspase-3 (#9661S, RTU, Cell Signaling) were used. DAB (Zymed) was used as substrate for peroxidase. Slides were counter-stained with hematoxylin and coverslipped using glycerin.

## Gene expression profiling of PDXs

DNA microarrays were used to define and compare the transcriptional profiles of PDXs treated with both Sal/JQ1 ($N = 3$) and their respective untreated controls ($N = 3$). Total RNA was extracted from PDX using the RNeasy Micro Kit (Qiagen), and quality was tested using an Agilent Bioanalyzer. Microarray experiments were done as recommended by the manufacturer (Affymetrix, Thermo Fisher) from 100 ng RNA for each sample using the Affymetrix GeneChip HuGene 2.0 ST arrays. Expression data were analyzed by RMA in R using Bioconductor and associated packages. The incidence of combined versus control treatment for each of the 11 subnetworks was assessed by computing a metagene-based score defined for each sample by the mean of subnetwork-associated gene expression. Each score was then compared between the Sal/JQ1-treated PDXs and the control ones using a logistic regression analysis with the

*glm* function in R. False discovery rate (FDR) was applied to correct the multiple testing hypothesis: The significant genes were defined by a FDR-adjusted *P*-value < 5%.

## Gene expression profiling of breast cancer data sets

We gathered clinicopathological and mRNA expression data of breast cancer samples from 36 public data sets. Data were collected from Gene Expression Omnibus (GEO, NCBI), ArrayExpress, and TCGA database (Dataset EV3). Samples had been profiled using whole-genome DNA microarrays (Affymetrix, Illumina or Agilent) and RNA-Seq (Illumina). The complete data set contained 10,233 samples, including 8,982 primary breast cancer samples of which 2,578 non-metastatic had available survival data. Data analysis required pre-analytic processing. First, we normalized each DNA microarray-based data set separately, by using quantile normalization for the available processed Agilent data, and Robust Multichip Average (RMA) (Irizarry *et al*, 2003) with the non-parametric quantile algorithm for the raw Affymetrix data. Normalization was done in R using Bioconductor and associated packages. Then, we mapped hybridization probes across the different technological platforms. We used SOURCE (SOURCE. The Stanford Online Universal Resource for Clones and ESTs. http://smd.stanford.edu/cgi-bin/source/sourceSearch) and NCBI EntrezGene (NCBI Entrez Gene. National Center for Biotechnology Information. http://www.ncbi.nlm.nih.gov/gene/) to retrieve and update the Agilent annotations, and NetAffx Annotation files (NetAffx Annalysis Center. http://www.affymetrix.com/analysis/index.affx) for the Affymetrix annotations. The probes were then mapped according to their Entrez-GeneID. When multiple probes represented the same GeneID, we retained the one with the highest variance in a particular dataset. Meta-subnetworks from interactome analysis were evaluated at mRNA level using metagene-based approach with respective gene lists. With a natural cut-off of 0, samples of each public data set were classified as high or low level for each of the selected meta-subnetworks.

## Statistical analysis

GraphPad Prism 5.0 was used for data analysis and imaging. Results are presented as the mean ± SD for at least three repeated independent experiments. To investigate associations among variables, univariate analysis was performed using non-parametric Wilcoxon rank sum test, chi-square test, or Student's *t*-test when appropriate. Extreme limiting dilution analysis (http://bioinf.wehi.edu.au/software/elda/) was used to evaluate bCSC frequency. A composite score (K) defining the drug effect on ALDHbr cell proportion was computed as the linear regression coefficient between untreated condition and a target dose where ALDHbr proportion decrease/increase of 50% compared to the control. The target dose was predicted using a 3$^{rd}$ degree polynomial regression model based on dose effect measurements for each drug and on each cell lines. K values close to 0 indicate resistance or no effect, whereas negative and positive one indicated, respectively, a decrease or an increase of the ALDHbr cell proportion. To compute K-score, drug doses were log10-transformed. For the two-drug combination screening, a 8 × 8 checker-board matrix format was used to identify synergy effect. The combination index (CI) for all dose associations was calculated using combenefit

### The paper explained

**Problem**

Breast cancer is a major health problem in developed countries. Although a significant improvement in the management of the disease, breast cancer remains the first cause of death by cancer in women. The breast tumor is a complex ecosystem of cells where a cell subpopulation, the so-called cancer stem cells (CSCs), seems to orchestrate tumor progression and therapeutic resistance. The discovery of CSCs provides an explanation for why cancer may be so difficult to cure and suggests new therapeutic strategies. If these cells are the tumor root, then they are the cells to be killed.

**Results**

In this paper, we performed a genome-wide RNA interference screen, an unbiased approach, to unreveal molecular mechanisms regulating breast CSCs-fate. We identified a panel of chemical compounds targeting these different pathways and validated the combination of salinomycin and JQ1 as the more efficient therapeutic association to target the breast CSC population. Using patient-derived xenograft, we demonstrated that salinomycin/JQ1 combination induces a depletion of the tumorigenic cells. Thus, we provide a new therapeutic approach to target the bCSCs population and prevent breast tumor progression.

**Impact**

A new generation of clinical trials is currently enrolling patients to validate CSC self-renewal process as a clinically relevant therapeutic target. This study offers a new anti-CSC therapeutic strategy that could lead to more successful control of the disease development and prevent treatment resistance.

tool based on the Loewe mathematical model (Di Veroli *et al*, 2016). To define significant drug dose combinations, we used a "neighborhood *Z*-score" combined with bootstrap resampling. Briefly, z-score using mean and standard deviation was computed for each CI and its closest neighbors onto the $8 \times 8$ matrix. Resulting *Z*-scores were compared to ones obtained after resampling CIs within 1,000 iterations to computed *P*-values. Drug combinations with a positive *z*-score and with a *P*-value < 0.05 were retained to have significant synergy. Relapse-free survival (RFS) was calculated from the date of diagnosis until the date of relapse. Follow-up was measured from the date of diagnosis to the date of last news for event-free patients. Survivals were calculated using the Kaplan–Meier method, and curves were compared with the log-rank test. Multivariate analysis was done using a Cox regression analysis where the best multivariate prognostic model was obtained by minimizing the Akaike information criterion (AIC) in a stepwise algorithm using both forward and backward directions. All statistical tests were two-sided at the 5% level of significance. Statistical analysis was done using the survival package (version 2.30) in the R software. We followed the reporting REcommendations for tumor MARKer prognostic studies (REMARK criteria) (McShane *et al*, 2005).

**Expanded View** for this article is available online.

## Acknowledgements

We express our gratitude to the "Fondation ARC" for supporting the acquisition of our cell sorter. This study was supported by Inserm, Institut Paoli-Calmettes, INCa (INCa_5911), SIRIC Marseille (INCa-DGOS-Inserm 6038), and the "Ligue National Contre le Cancer" (Label DB). VA has been supported by a fellowship from the "Ligue National Contre le Cancer". Thanks are due to the CRCM flow cytometry and animal core facilities.

## Author contributions

AA performed most of the experiments, analyzed the data, and prepared the original draft of the manuscript. CR and GB developed HTS-net and performed the interactome/regulome analysis. GP and SL performed the RNAi screen. VA, JW, CL-S, and AB performed *in vitro* experiments and analysis with chemical compounds. AG, EJ, MM, and RC performed *in vivo* experiments. PF performed biostatistical analysis. MK, YC, FB, DB, and HD reviewed and edited the manuscript. CG and EC-J conceived and supervised the project and wrote the paper. All authors discussed the results and commented on the manuscript.

## Conflict of interest

The authors declare that they have no conflict of interest.

## For more information

http://mobylehome.marseille.inserm.fr/cgi-bin/portal.py#forms::HTS-Net
http://bioinf.wehi.edu.au/software/elda/

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
