## [Review Process File · EMBO Molecular Medicine]

Title: A genome-wide RNAi screen reveals essential therapeutic targets of breast cancer stem cells

Authors: Abir Arfaoui, Claire Rioualen, Violette Azzoni, Guillaume Pinna, Pascal Finetti, Julien Wicinski, Emmanuelle Josselin, Manon Macario, Remy Castellano, Candi Léonard-Stumpf, Anthony Bal, Abigaëlle Gros, Sylvain Lossy, Maher Kharrat, Yves Collette, Francois Bertucci, Daniel Birnbaum, Hayet Douik, Ghislain Bidaut, Emmanuelle Charafe-Jauffret and Christophe Ginestier.

Review timeline:

Submission date:	13 th October 2018
Editorial Decision:	21 st November 2018
Revision received:	14 th June 2019
Editorial Decision:	10 th July 2019
Revision received:	13 th July 2019
Accept:	7 th August 2019

Editor: Lise Roth

Transaction Report:

1st Editorial Decision

21st November 2018

Thank you for the submission of your manuscript to EMBO Molecular Medicine. We have now heard back from the three referees whom we asked to evaluate your manuscript.

As you will see from the reports below, while they all mention the interest of the study, they also raise substantial concerns on your work, which should be convincingly addressed in a major revision of the present manuscript. Our cross-commenting exercise helped crystallizing the major points to be addressed to fully support the conclusions and the translational relevance of the findings:

- 1/ While it is not realistic to add several other cellular models *in vivo*, control cell lines from different breast cancer subtypes should be tested *in vitro* to show the specificity of the therapy. Testing one additional TNBC cell line model and one additional PDX model *in vivo* would further strengthen the results.
- 2/ The effects of the therapy on the primary tumor as well as on metastasis (upon primary tumor removal and/or upon experimental metastasis) should be shown.
- 3/ The reviewers criticized the over-reliance on a unique stem cell marker, and this point should be addressed.
- 4/ The text (abstract, results, discussion) should be tuned down to fully and accurately reflect the experiments and the results.

Addressing the reviewers concerns in full (above points as well as other reviewers' comments) will be necessary for further considering the manuscript in our journal. Still, revising the manuscript according to the referees' recommendations appears to require a lot of additional work and experimentation. I am unsure whether you will be able or willing to address those and return a revised manuscript within the 3 months deadline. On the other hand, given the potential interest of

the findings, I would be willing to consider a revised manuscript with the understanding that acceptance of the manuscript would entail a second round of review as some claims were not fully evaluated from the current data due to limitations highlighted by the referees.

REFEREE REPORTS

Referee #1 (Remarks for Author):

Using a genome-wide RNA interference screen in a breast cancer line, the authors identified genes regulating breast cancer stem cells (defined as ALDH1^{high}), and uncovered different regulatory networks controlling CSC. They use panels of previously described inhibitors to interfere with these regulatory networks and found that a combination of salinomycin and JQ1 is the most efficient combination to deplete the CSC population in vitro and in xenografts. While this study is of potential interest for readers of EMBO Molecular Medicine, there are some issues that need to be addressed before publication.

Major comments

-While the effect of the combination of salinomycin and JQ1 on the number of ALDH1^{high} cells and CSC (defined as tumor initiating cells upon transplantation) is convincing, the relevance as a therapeutic tool is missing, as no or limited effect on tumor size is observed. Does this drug combination impact on metastasis (if the primary tumor is removed, and upon IV injection of tumor cells)? Does this drug combination work in synergy with conventional anti-cancer therapy? Besides the number of ALDH1 cells, how is the primary tumor affected by the CSC depletion? Is a change in differentiation, apoptosis or necrosis observed?

-Previous work from the authors has shown that repertaxin targets CSC and blocks metastasis formation. A comparison and discussion of the salinomycin/JQ1 and the repertaxin effects, alone or in combination, would be of interest.

-Only 2 xenografts models are used, which clearly is not enough to reflect breast cancer heterogeneity. From which breast cancer type are they derived? Xenografts from each breast cancer subtype should be tested and analysed separately, as they might respond differently to the therapy.

Minor comments

-The abstract mention that treatment of xenografts with the drug combination "resulted in long-term impairment of tumor growth", while the results rather show that "the drug combination treatment had no or limited effect on PDXs growth" in the timeframe of the experiments. The abstract should reflect the results and this overstatement should be corrected.

-5 different breast cancer cell lines are used to test the anti-CSC compounds, to reflect diversity of breast cancer. It would be helpful to explicit in the text from which breast cancer subtype is derived each of these cell lines.

-The result section is hard to follow as the text describes drugs from groups without mentioning their protein target.

-The rationale for using 4 of the 5 cell lines initially used in figure 4 should be explained.

-Figure 1E. The legend mentions Hoechst staining in blue, which is missing in the figure.

-Figure 3A. The table is missing subnetwork 11.

-Figure 3B. The description of the 12 conditions shown in the figure is missing.

-Figure 5C, D. The legend should clearly mention if the number of ALDH1 high cells is normalized only to the number of ALDH1 high cells in control or also to the size of tumor?

Referee #2 (Comments on Novelty/Model System for Author):

Improving on this model system would require similar RNAi screens on more than one breast cancer cell line and or the use of PDX tumors.

Referee #2 (Remarks for Author):

In this manuscript, Arfaoui et al provide evidence that a genome-wide RNA interference (RNAi) screen regulates breast cancer stem cell (bCSC) fate. Utilizing SUM159PT as the cellular model and an ALDEFLUOR based assay, the bCSC are identified as ALDH^{bright}. The authors identified a

gene hit-list that significantly decreases the CSC proportions and a second gene hit list whose silenced genes led to an expansion of bCSCs. The authors suggest that the gene hit-lists are genes which regulate bCSC-fate. A series of networks identified by interactome/regulome analysis is suggested by the authors to be relevant to bCSC fate. Subsequently a targeted-pool of 33 RNAi from the library were used to screen multiple breast cancer cell lines altering CSC activity and directly compared and contrasted to a chemical screen in the various breast cancer cell lines. Based on the above the authors determined that mifepristone, salinomycin and JQ1 targeted bCSC activity. In vitro exposure of salinomycin and JQ1 synergized altering stem cell activity. In vivo, xenograft models impaired tumor growth in PDX models and in limited dilution assays with PDX cells. Relevance to human disease was shown by the analysis of public available dataset on recurrence free survival rates were poorer when the network signature was present. The authors suggest that the networks arising from the RNAi-screen are drivers responsible for tumor CSC evolution arising from chemoresistance.

General concerns of the text and results description were found to be sparse and often lacking in accurate details that match the text to what was shown in the figures requiring at times to turn to the legends for details and at times vital information was missing. The text should stand alone without going to figure legend and then find out value information is missing. The manuscript text should be able, for general readership not experts on RNAi screen technologies or knowledge of the author's previous publications.

Another general concern is the time frame experiment with the PDX models in vivo. The authors acknowledge that they are conducted so as to make the argument that this specific-modeling is consistent with what would be expected of CSCs, as it is responding to therapy and would be undergoing clonally evolving due to therapy. There is no confirmation and/or secondary validation experiments at the molecular level that demonstrate that the drugs are indeed affecting CSC networks as they evolve to resist the drugs. No validation is shown that the RNAi-signature networks are working in vivo to promote chemoresistance.

The authors need to address the general concerns and the major/minor points.

Major Points:

1. Figure 1 A-B are generally easy to follow along.
2. The volcano plot, which represents the gene hit list 1 (green down) and hits list 2 (red genes up), needs more than just ALDH1 highlighted. Then one can follow the analysis of targets chosen (EP300 EZH2 are the red) and (ATG5 NR3C1 and MED09 green). Because there is no mention of EP300 and EZH2 etc... in the text or the legends. Confusing to read.
3. Supplemental Figure 1 was an exhausting excel sheet. Was this hit-list 1 and hit -list2? Not clear. Supplemental description was very poor. Summary tables for clarification of those gene hit-lists.
4. Is the software developed by authors (Rioualen et al 2017) freely available for independent verification analysis by others in the field?
5. Fig2. The five networks identified from the results of Figure 1. Are the authors implying that this is the driver of clonal evolution and this is due to resistance? This point was lost in text please clarify.
6. Fig3A. The text is not clear on why you chose the EZH2 inhibitor tazemetosat? EZH2 is not the "hub" of network core, it is KNA2, what is it because EZH2 was one of the gene hit list that went up? And there is no drug to target for the hub, KNA2?
7. Fig 3B would benefit by sublabels stipulating that we are looking at 33 siRNA as the text says or as the legend says 12 siRNA pools 3 each (to me it add up to is 36 targeted RNAi) please correct. Moreover which were targeted genes? Not clear from the text or the legend. I can image it the above genes that was shown in figure 1.
8. Fig 3B multi-rounds of therapy on the SFE, self-renewal capacity must be considered. As it is now well established the SFE are influenced by the levels of ECADHERIN for a given breast cancer cell line and causing aggregation of cell to form spheres.
9. Fig 3C-D it is not clear why the authors using control EZH2 inhibitor on the SUM159PT then JQ1 on SUM149, why the switch in the cells? What about Salinomycin?
10. The JQ1 and Salinomycin effects specific only to TNBC cells lines? What about ER+ HER2+ normal epithelial cells etc... controls are needed?

11. Fig 3C the authors measuring the effects of DEAB as the inhibitor for human ALDH co-enzymes. This is not well described at all in either the text or the figure legends for some readers that may not know this assay it would be difficult to assess what is being shown.
12. Supplemental 1 figure dosage p values are lacking for each cell lines drug used dosages.
13. Figure 4 Isobole curve analysis for 24-96h are needed to determine appropriate drug interactions and determining synergistic, additive or combinatorial.
14. What is the mechanism of the action on the networks for JQ1 and Salinomycin alone and or combination as their mechanism varies?
15. Fig5 is strong but lacks verification that the intended networks were truly affected in vivo by drug therapies as suggested by the authors.

Minor Points

1. The RFS are statistically significant for TNBC. But there was no mention of the HR values and the discussion does not address this point. Additionally explanation as to why we see significant changes but that they are marginal changes relative small percentages?
2. There was no discussion what is the relevant dosages used in mice to potential translate to human studies that would target CSC. Are you exceeding levels that are not tolerable in humans, otherwise no impact.

Referee #3 (Remarks for Author):

The manuscript by Abir Arfaoui et al., entitled " A genome-wide RNAi screen reveals essential therapeutic targets of breast cancer stem cells" describes an integrated interactome/regulome analysis based on a genome-wide RNA interference screen in a cell-based model system of basal-type breast cancer, the SUM159 cell line. The ultimate goal of this study is the characterization of genes/pathways that might constitute suitable targets for a rationale anti-cancer stem cell (CSC) therapy in breast cancer. By this approach, the Authors identified potential underpinnings of CSC phenotypes, which were subsequently validated through in vitro and in vivo functional studies as suitable targets for pharmacological intervention using a number of available compounds. An important outcome of these analyses is the identification of at least two drugs, salinomycin, previously described as a selective anti-CSC drug, and JQ1, which can be used alone or in combination to target CSCs.

The manuscript addresses important questions in the field of therapeutic management of breast cancer, first and foremost the necessity to target CSCs to achieve definitive tumor eradication, as apposed to currently available strategies that in most cases exert a predominant effect on the bulk tumor population. In this contention, a highlight of this work is the use of in vitro and in vivo functional assays in an attempt to dissect the effects of the selected compounds on the CSC subfraction, rather than on the bulk tumor population. This approach is in keeping with the emerging idea that functional assays that interrogate CSC biology should be routinely implemented in the evaluation of efficacy of anti-tumor drugs. Notwithstanding these important and valuable aspects, there are several pitfalls to this work that detrimentally affect its overall significance and clinical impact at this time.

General comment

One major point of concern is the use of the ALDH-based technique as the sole phenotypic assays to "count" CSCs throughout the study. Indeed, while the use of ALDH in combination with the basal-like SUM159 cell line appears to be a suitable strategy, entirely justified for the execution of a high-throughput analysis, there is a sort of over-reliance on ALDH as a "universal" approach to measure CSCs across the different cell lines and patient-derived cells (the molecular subtypes of patient is not described) in the subsequent validation studies. At least in the case of basal-type breast cancer, other CSC markers can be also exploited (for instance the CD44/CD24 configuration) which might have aided a better definition of the effects of the compounds on the CSC population. Notwithstanding this, the combined SUM159/ALDH approach revealed to be a suitable strategy to validate the effects of the identified genes on the putative ALDH-positive CSC fraction, considering

the good correspondence between ALDH-positive cells and sphere-forming efficiency *in vitro* (at least in some experiments). In this regard, a major point of weakness is the absence of evidence that some of these genes are also functionally implicated in determining variations of the CSC pool *in vivo*. Such evidence would have strongly reinforced the significance of pharmacological studies *in vivo*, based on the rationale that those genes/pathways are the actual molecular target(s) for the selected drugs. Differently said, one would expect that the perturbation of the identified genes/pathways, singly or in combination, would recapitulate the effects of the pharmacological treatments, which would provide definitive preclinical proof-of-evidence of the suitability of using the candidate compounds as anti-CSC treatment(s) in the clinical setting.

Another important point concerns the 80% cut-off for proliferative block/apoptosis effects used to identify candidate hits in the RNAi screen in the bulk SUM159 population. If the purpose of the study was to identify genes/pathways with a selective effect on CSCs, which are hold to be predominantly quiescent/slow-proliferating/apoptosis-resistant, rather than an effect of the bulk population of actively proliferating progenitors, it would have been more appropriate to evaluate the differential apoptosis/proliferative effects on the purified ALDH-positive and -negative subfractions.

SUM159 cells are basal-type, as apparently are all the other cell lines utilized in the study, (while the molecular and clinical characteristics for the two patients appear not be reported). In this regard, if it is true that basal-type breast cancers are among the most biologically aggressive breast cancer subtypes, they also represent a minority of breast cancers (where the great majority is represented by luminal breast cancers which are also presumably driven by CSCs). Assessing the implication of the identified genes/pathways and related targeted therapies in other breast cancer subtypes would have greatly strengthened the overall biological and clinical impact of the study, in light of the emerging idea that breast CSCs might share molecular and phenotypical homologies regardless of the heterogeneity that distinguishes the different molecular subtypes at the bulk population level.

The putative anti-CSC drugs identified, salinomycin - which has been already described as an anti-CSC drug -, JQ1 - which should target the sub-network mediator complex - and mifeprestone - that targets NR3C1- appear to show highly variable, sometimes not reproducible, effects when their efficacy *in vitro* (ALDH/sphere-forming assay) is compared to their efficacy *in vivo*. Moreover, only salinomycin shows a consistent effect when used alone or in combinatorial treatment, while all the other drugs display a modest, if any, effect when used alone.

Furthermore, accurate evaluation of the growth kinetics of xenografts *in vivo* appears to show a marked anti-proliferative effect in the bulk tumor population. This point should be carefully addressed with an accurate analysis of the rate of apoptosis and proliferation in the treated bulk xenograft population. For instance (Fig. 5), the effects of salinomycin and JQ1 on the CSC content, and on the growth rate of the primary xenograft appear to be inconsistent compared to each other. In addition, the absence of effect of salinomycin on the growth of the primary xenograft (which is expected of a selective anti-CSC therapy) does not correspond to a consistent action of salinomycin in reducing the content of ALDH-positive CSCs; even more concerning, JQ1 appears to be endowed with a remarkable anti-tumor growth activity (which is of the same order of magnitude of the combined salinomycin+JQ1 treatment) which is not accompanied by any substantial effect of this drug on the ALDH-positive fraction. This suggests that the site of action of JQ1 might be at the level of the bulk population of progenitors, rather than hitting the CSC population. Again, while the effects of the different treatments on the growth of the secondary xenografts are remarkable, a thorough analysis of the apoptosis/proliferative effects in the bulk tumor population would aid understanding of the mechanism/site of action of the different treatments. Therefore, in light of the different caveats and pitfalls of the above *in vivo* studies, the conclusion that these drugs represent selective anti-CSC treatments remains questionable.

Major points:

Fig. 3.

It would be useful to show, in addition to the sphere-forming efficiency in panel B, a table reporting the actual average size of the spheres (expressed as number of cells/sphere) alongside measurement of the actual proliferative rate (for instance, cell cycle analysis; also in panel C/D) in the bulk sphere population, with the ultimate goal to demonstrate the differential effects of the treatments on the putative CSC (ALDH-pos) vs. progenitor (ALDH-neg) population. A serial propagation of spheres

(at least a second generation) would definitively aid addressing the selective anti-CSC effect of the various treatments, at least for selected genes/drugs.

The effect of mifepristone and JQ1, shown in Fig. G is really modest, compared to salinomycin, which questions their actual efficacy in this assay. As previously stated, also in this analysis, the analysis of the actual sphere size and of the apoptosis/proliferative rate would probably be instrumental to better highlighting the intrinsic mechanism/cellular site of action of these compounds.

Fig. 4.

Results depicted in this figure would probably be better represented by a synoptic table showing the selected final optimal concentrations of the different drugs, their effects in terms of sphere-forming efficiency and the intrinsic characteristics of the different cell lines (subtypes).

The experiments depicted in panel C lacks fundamental controls (not-treated control, salinomycin alone at 30 and 500nM) which are required to understand whether this drug is already effective as a standalone treatment, or whether a combinatorial regimen is required to maximize its efficacy, especially in light of the relatively modest effects that JQ1, and to some extent also salinomycin, appear to exert when used in mono-therapy. This would also aid understanding whether the different drugs are hitting different CSC pools (or different molecular workings), considering their synergic effects when used in combination, or whether their combination simply results in off-targets effects (toxicity?).

In these experiments, it also concerning that the remarkable reduction (~8-10-fold) observed in the percentage of ALDH-positive cells upon inhibitor treatments (panel C) does not reflect into an equivalent effect in the sphere-forming assay. Once again, accurate dissection of the possible effects on proliferation/apoptosis in these assays, accompanied by second generation sphere formation assays, would help addressing these points. It is also possible that ALDH is not sufficient to "read" the selective effects on the CSC population, which might be better interpreted with different approaches to CSC purification (i.e. CD24/CD44-based sorting, if amenable).

Fig. 5.

This figure depicts results of possible translational relevance, and as such represents a major added value of the study. Increasing the number of patients including different breast cancer subtypes would greatly increase the impact of these results, also considering the remarkable variations observed in the analysis of the two patients. Are these patients p53 wild-type or mutated? More in general, what are the intrinsic characteristics of these two patients?

A thorough analysis of the effects of the drug treatments (proliferation/apoptosis) is also required, as already suggested above. It is also not very well described when the treatments were started. Again, as previously stated, salinomycin and JQ1 appear to have variable, inconsistent effects on the CSC content, when the ALDH method is compared to the sphere-forming assay.

In the limiting dilution transplantation assays, while some treatments appear to have a more remarkable effect than others, the combinatorial treatments result in a dramatic inhibition of the tumor growth, which raises concerns about the selective anti-CSC effects of these treatments. This point should be carefully addressed by corroborating these studies with functional (sphere-forming efficiency, ALDH assay) and in situ (detection of apoptosis/proliferation rate) analyses of the tumor outgrowths. Taken at a face value, results from in vivo studies appear to indicate that a combinatorial treatment is not strictly required to achieve a substantial effect. Towards assessing the preclinical relevance of these studies, it would also be useful to evaluate the combination of (at least one) candidate drugs with standard chemotherapy, to observe whether the selective anti-CSC action of one of these compounds prevent tumor growth relapse after chemotherapy discontinuation (as it would be expected in case of intrinsic CSC resistance to chemotherapy).

Additional points:

In some parts, the manuscript does not read well. The definition "breast CSC" or the acronym bCSC are inconsistently used throughout the manuscript.

Answer to reviewers' comments:

Referee #1 (Remarks for Author): Using a genome-wide RNA interference screen in a breast cancer line, the authors identified genes regulating breast cancer stem cells (defined as ALDH1^{high}), and uncovered different regulatory networks controlling CSC. They use panels of previously described inhibitors to interfere with these regulatory networks and found that a combination of salinomycin and JQ1 is the most efficient combination to deplete the CSC population in vitro and in xenografts. While this study is of potential interest for readers of EMBO Molecular Medicine, there are some issues that need to be addressed before publication.

Major comments

1. While the effect of the combination of salinomycin and JQ1 on the number of ALDH1^{high} cells and CSC (defined as tumor initiating cells upon transplantation) is convincing, the relevance as a therapeutic tool is missing, as no or limited effect on tumor size is observed. Does this drug combination impact on metastasis (if the primary tumor is removed, and upon IV injection of tumor cells)?

We thank the reviewer for his/her helpful comments.

We agree with the reviewer that an impact of salinomycin/JQ1 drug combination on metastasis formation will grandly reinforce the clinical relevance of this therapeutical approach. Thus, we have now performed a metastasis formation assay upon IV injection of tumor cells isolated from two of our PDX models expressing the luciferase. At 24 hours after IV injection, mice were treated daily with vehicle, salinomycin, JQ1, or the drug combination. After 14 days of treatment, mice were sacrificed and metastatic burden was determined by photon flux emission. We obtained similar results in both PDX models with a significant reduction of metastasis formation in mice treated by the drug combination compared to the vehicle-treated mice. These results are now reported in Figure 5 (J-L) and in Supplementary Figure 6

2. Does this drug combination work in synergy with conventional anti-cancer therapy?

We agree with the reviewer that it is important to associate our drug combinations targeting the CSC population with standard chemotherapy more efficient on bulk cell population. We have now tested this association (using docetaxel as standard chemotherapy) in two different cell line models. We obtained similar results for both models with JQ1, Sal, or JQ1/Sal treatments that prevent CSC selection induced by docetaxel treatment. These results are now reported in Supplementary Figure 4. The next step will be to develop a preclinical assay in PDXs testing the different regimen (concomitant, sequential) of tri-therapy

(JQ1/Sal/Docetaxel). This step will be important to translate our therapeutic approach into clinics. However, we think that it is beyond the scope of the present manuscript.

3. Besides the number of ALDH1 cells, how is the primary tumor affected by the CSC depletion? Is a change in differentiation, apoptosis or necrosis observed?

The reviewer raised a critical question. Changes in bCSC proportion may be explained by different scenarios such as, preferential or selective bCSC apoptosis, modification of proliferation rate of one or both cell subpopulations (ALDH^{br} and ALDH⁻), and increased bCSC differentiation, or a combination of each. To evaluate how treatments affect bCSC proportion we have first measured the proportion of apoptotic cells in each cell subpopulations following treatments (JQ1, SAL, JQ1/SAL). We did not observed a significant increase of apoptotic cells in any of the two cell subpopulations following treatments. These results are now reported in Figure 4E. To evaluate the impact of the treatment on the cell proliferation and on the bCSC differentiation potential we developed a lineage tracing system using an engineered SUM 159 cell line (see new Figure 4F). Using this model we can follow the non-CSC (ALDH^{neg}-RFP⁺ cells) and CSC populations (ALDH^{pos}-BFP⁺ cells) progenies under treatment conditions. We did not observed any modifications of the RFP⁺/BFP⁺ cells ratio in the treated conditions compared to the control, indicating that treatments did not induce a differential proliferation rate in any of the two cell subpopulations. However, we did observe a significant reduction of the ALDH^{br} cell proportion in the CSC progenies (BFP⁺ cells) under treatments compared to the control (see new Figure 4G-I). These results suggest that JQ1 and Sal treatments mainly reduced the CSC proportion by promoting CSC differentiation.

Previous work from the authors has shown that repertaxin targets CSC and blocks metastasis formation. A comparison and discussion of the salinomycin/JQ1 and the repertaxin effects, alone or in combination, would be of interest.

We have indeed identified repertaxin as a CSC targeting drugs able to block metastasis formation. Repertaxin is currently involved in an international clinical trial for metastatic breast cancer (FRIDA, NCT02370238). As answered previously, we have now performed a metastasis formation assay (see comments above). We really do think that only a well-designed clinical trial will be able to evaluate which CSC targeting drug is more beneficial for patients cure and we will definitely designed new studies to answer this question.

4. Only 2 xenografts models are used, which clearly is not enough to reflect breast cancer heterogeneity. From which breast cancer type are they derived? Xenografts from each breast cancer subtype should be tested and analysed separately, as they might respond differently to the therapy.

All our in vitro experiments have been performed on a panel of five breast cancer cell lines (BCLs) representing the breast cancer molecular diversity (1 luminal BCL, 2 Basal BCLs, and

2 Mesenchymal BCLs). Similar results have been obtained for all these BCLs. Concerning our *in vivo* experiments, we have focused our study on the triple-negative breast cancers that are still facing unmet medical needs. To reinforce our study, we have now tested one additional PDX model of triple-negative breast cancer. In all three tested PDX models, we obtain similar results with a significant effect of the salinomycin/JQ1 drug combination on the CSC population, limiting tumor progression. These results are now reported in Figure 5 and in Supplementary Figure 6. The description of our PDX models has been added in the Material and Methods section.

Minor comments.

1. The abstract mention that treatment of xenografts with the drug combination "resulted in long-term impairment of tumor growth", while the results rather show that "the drug combination treatment had no or limited effect on PDXs growth" in the timeframe of the experiments. The abstract should reflect the results and this overstatement should be corrected.

We agree with the reviewer that this point was indeed overstated. We modified the text as follows: "Treatment of primary breast cancer xenografts with this combination reduces the tumor-initiating cell population and limits metastatic development".

2. 5 different breast cancer cell lines are used to test the anti-CSC compounds, to reflect diversity of breast cancer. It would be helpful to explicit in the text from which breast cancer subtype is derived each of these cell lines.

The molecular subtypes of each breast cancer cell lines used in this study are now reported in the material and methods section.

3. The result section is hard to follow as the text describes drugs from groups without mentioning their protein target.

The manuscript was modified for improved clarity and readability of the results section.

4. The rationale for using 4 of the 5 cell lines initially used in figure 4 should be explained.

The effect of the panel of drugs on the CSC population from Br-Ca-MZ01 cell line was limited. Thus, we decided to focus our combination assay on the four cell line models with the strongest response to treatments. We have now modified the text as follow: "We computed the individual dose-response matrixes testing pairwise combinations of eight doses of the selected drugs on the four BCLs with the strongest response to treatments with single drugs".

5. Figure 1E. The legend mentions Hoechst staining in blue, which is missing in the figure.

We made the correction

5. Figure3A. The table is missing subnetwork 11.

We did not identified actionable target with corresponding inhibitors for subnetwork-11

6. Figure 3B. The description of the 12 conditions shown in the figure is missing.

We made the corrections

7. Figure 5C, D. The legend should clearly mention if the number of ALDH1 high cells is normalized only to the number of ALDH1 high cells in control or also to the size of tumor?

The ALDHbr cell proportions in the drugs-treated tumors are normalized only to the number of ALDHbr cell proportions detected in the vehicle-treated tumors. We have now mentioned this point in the figure legend.

Referee #2 (Comments on Novelty/Model System for Author): Improving on this model system would require similar RNAi screens on more than one breast cancer cell line and or the use of PDX tumors.

Referee #2 (Remarks for Author): In this manuscript, Arfaoui et al provide evidence that a genome-wide RNA interference (RNAi) screen regulates breast cancer stem cell (bCSC) fate. Utilizing SUM159PT as the cellular model and an ALDEFLUOR based assay, the bCSC are identified as ALDHbright. The authors identified a gene hit-list that significantly decreases the CSC proportions and a second gene hit list whose silenced genes led to an expansion of bCSCs. The authors suggest that the gene hit-lists are genes which regulate bCSC-fate. A serie of networks identified by interactome/regulome analysis is suggested by the authors to be relevant to bCSC fate. Subsequently a targeted-pool of 33 RNAi from the library were used to screen multiple breast cancer cell lines altering CSC activity and directly compared and contrasted to a chemical screen in the various breast cancer cell lines. Based on the above the authors determined that mifepristone, salinomycin and JQ1 targeted bCSC activity. In vitro exposure of salinomycin and JQ1 synergized alerting stem cell activity. In vivo, xenograft models impaired tumor growth in PDX models and in limited dilution assays with PDX cells. Relevance to human disease was shown by the analysis of public available dataset on recurrence free survival rates were poorer when the network signature was present. The authors suggest that the networks arising from the RNAi-screen are drivers responsible for tumor CSC evolution arising from chemoresistance.

General concerns of the text and results description were found to be sparse and often lacking in accurate details that match the text to what was shown in the figures requiring at times to turn to the legends for details and at times vital information was missing. The text should stand alone without going to figure legend and then find out value information is missing. The manuscript text should be able, for general readership not experts on RNAi screen technologies or knowledge of the author's previous publications.

We thank the reviewer for his/her helpful comments.

We apologize for the troubles encountered by the reviewer to read properly the manuscript. We have now carefully answered all its major/minor points to improve the readability of our manuscript text.

Another general concern is the time frame experiment with the PDX models in vivo. The authors acknowledge that they are conducted so as make the argument that this specific-modeling is consistent with what would be expected of CSCs, as it is responding to therapy and would be

undergoing clonally evolving due to therapy. There is no confirmation and or secondary validation experiments at the molecular level that demonstrate that the drugs are indeed affecting CSC networks as they evolve to resist the drugs. No validation is shown that the RNAi-signature networks are working in vivo to promote chemoresistance.

The reviewer raised an important point. To answer his/her comment we have now performed a gene expression profiling analysis comparing PDXs (CRCM404 and CRCM434) treated with Salinomycin/JQ1 combination to their respective vehicle-treated tumors. Using metagene-based score calculated for each subnetworks (defined in Figure 2), we were able to show that only subnetwork-6 (mediator complex) and subnetwork-9 (Autophagy) presented a significant down-regulation in tumors treated with Salinomycin/JQ1 combination compared to the vehicle-treated tumors. These observations confirm that Salinomycin/JQ1 combination specifically targets their related subnetworks. These results are now reported in Figure 4M and Supplementary Figure 7C.

The authors need to address the general concerns and the major/minor points.

Major Points:

1. Figure 1 A-B are generally easy to follow along.
2. The volcano plot, which represents the gene hit list 1 (green down) and hits list 2 (red genes up), needs more than just ALDH1 highlighted. Then one can follow the analysis of targets chosen (Ep300 EZH2 are the red) and (ATG5 NR3C1 and MED09 green). Because there is no mention of EP300 and EZH2 etc... in the text or the legends. Confusing to read.

We modified the Figure 1F to highlight the different targets illustrated in Figure 1C-E

3. Supplemental Figure 1 was an exhausting excel sheet. Was this hit-list 1 and hit -list2? Not clear. Supplemental description was very poor. Summary tables for clarification of those gene hit-lists.

To clarify the readability of the Supplementary Table 1, we have now added two folders regrouping hits that either induce a decreased of the CSC proportion (named Hit-list 1) or induce an expansion of the CSC population (named Hit-list 2). Moreover we have modified the text and the supplementary Table 1 legend to clarify the description of this supplementary table.

4. Is the software developed by authors (Rioualen et al 2017) freely available for independent verification analysis by others in the field?

The HTS-net algorithm is freely available. We have now added the website address in the material and methods section (<http://mobyhome.marseille.inserm.fr/cgi-bin/portal.py#forms::HTS-Net>).

5. Fig2. The five networks identified from the results of Figure 1. Are the authors implying that this is the driver of clonal evolution and this is due to resistance? This point was lost in text please clarify.

The functional mapping using HTS-net algorithm allows the detection of networks that regulate the breast CSC proportion. It may have an impact on clonal evolution but our study does not explore this question. Concerning the therapeutic resistance, we showed that patients with a high expression of subnetworks-6 and -9 are associated with a worse relapse-free survival compared to patients with low expression level, suggesting a potential effect on therapeutic resistance. These observations are described in Figure 6.

6. Fig3A. The text is not clear on why you chose the EZH2 inhibitor tazemetosat? EZH2 is not the "hub" of network core, it is KNA2, what is it because EZH2 was one of the gene hit list that went up? And there is no drug to target for the hub, KNA2?

To identify potentially actionable targets that may affect the breast CSC proportion, we selected, from the whole gene list that composed the 11 subnetworks, genes encoding proteins for which chemical inhibitors were available. We did not identify drugs targeting specifically KPNA2.

7. Fig 3B would benefit by sublabels stipulating that we are looking at 33 siRNA as the text says or as the legend says 12 siRNA pools 3 each (to me it add up to is 36 targeted RNAi) please correct. Moreover which were targeted genes? Not clear from the text or the legend. I can image it the above genes that was shown in figure 1.

We agree with the reviewer that this point was confusing. We developed a focused library of 33 siRNAs designed to inhibit the 11 genes/pathways and compared this effect to a pool of 3 scrambled siRNA. Thus, we end up with a total of 12 siRNA pools. To avoid any confusion we have now re-annotated Figure 3B and modified the figure legend.

8. Fig 3B multi-rounds of therapy on the SFE, self-renewal capacity must be consider. As it is now well established the SFE are influenced by the levels of ECADHERIN for a given breast cancer cell line and causing aggregation of cell to form spheres.

We agree with the reviewer and we have now performed secondary tumorsphere assays. We confirmed the impact of the treatment on the CSC population. These results are now reported in Figure 3H and Figure 4D.

9. Fig 3C-D it is not clear why the authors using control EZH2 inhibitor on the SUM159PT then JQ1 on SUM149, why the switch in the cells? What about Salinomycin?

Figure 3C-D shows representative examples of ALDHbr cell variation according to a drug ranging dose. All the detailed combinations of drugs and breast cancer cell models are presented in supplementary Figure 2. We have now modified the figure 3C-D legend to clarify this point.

10. The JQ1 and Salinomycin effects specific only to TNBC cells lines? What about ER+ HER2+ normal epithelial cells etc... controls are needed?

All our in vitro experiments have been performed on a panel of five breast cancer cell lines (BCLs) representing the breast cancer molecular diversity (1 luminal BCL, 2 Basal BCLs, and 2 Mesenchymal BCLs). Similar results have been obtained for all these BCLs. For our in vivo

experiments, we have focused our study on the triple-negative breast cancers that are still facing unmet medical needs.

Concerning the potential effect on normal cells, we did not detect major side effects in mice treated with JQ1 and salinomycin alone or in combination.

11. Fig 3C the authors measuring the effects of DEAB as the inhibitor for human ALDH co-enzymes. This is not well described at all in either the text or the figure legends for some readers that may not know this assay it would be difficult to assess what is being shown.

We made the clarification in the legend of Figure 3 and Figure 4.

12. Supplemental 1 figure dosage p values are lacking for each cell lines drug used dosages.

Supplementary Figure 1 represents the dose-response curves based on the cell viability of five BCLs following treatment with the panel of 15 drugs. Based on these curves, we determined each drug- and cell line-specific IC50 (dashed-green line). We modified the supplemental Figure 1 to clarify this point.

13. Figure 4 Isobole curve analysis for 24-96h are needed to determine appropriate drug interactions and determining synergistic, additive or combinatorial.

To determine synergy effect between two drugs we used Combenefit tool based on the Loewe mathematical model (Di veroli et al., Bioinformatics, 2016). Using this algorithm, we do not need 24h or 96h time points to accurately calculate drug interactions. To avoid any misunderstandings we removed the isobole curves from Figure 4. These curves have not been directly generated by the mathematical model which rely on the on the dose-response matrixes.

14. What is the mechanism of the action on the networks for JQ1 and Salinomycin alone and or combination as their mechanism varies?

We agree with the reviewer that deciphering the mechanistic role of JQ1 and salinomycin combination is a crucial point that will need to be further investigate. Although we have begun experiments to study the relationship between JQ1 and Salinomycin, we believe that it is beyond the scope of the present manuscript where we first identified this drug combination as potential new anti-CSC therapy.

15. Fig5 is strong but lacks verification that the intended networks were truly affected in vivo by drug therapies as suggested by the authors.

The reviewer raised an important point. To answer his/her comment we have now performed a gene expression profiling analysis comparing PDXs (CRCM404 and CRCM434) treated with Salinomycin/JQ1 combination to their respective vehicle-treated tumors. Using metagene-based score calculated for each subnetworks (defined in Figure 2), we were able to show that only subnetwork-6 (mediator complex) and subnetwork-9 (Autophagy) presented a significant down-regulation in tumors treated with Salinomycin/JQ1 combination compared to the vehicle-treated tumors. These observations confirm that Salinomycin/JQ1 combination specifically targets their related subnetworks. These results are now reported in Figure 4M and Supplementary Figure 7C.

Minor Points

1. The RFS are statistically significant for TNBC. But there was no mention of the HR values and the discussion does not address this point. Additionally explanation as to why we see significant changes but that they are marginal changes relative small percentages?

We have now added the HR values in the text.

2. There was no discussion what is the relevant dosages used in mice to potential translate to human studies that would target CSC. Are you exceeding levels that are not tolerable in humans, otherwise no impact.

There is currently no available literature about early phase clinical trials testing dose escalation for these two compounds (Sal and JQ1). However based on usual dose conversion between mice and human (Nair and Jacob, J Basic Clin Practice, 2016) our treatment regimens are in the standard dose range. Moreover, we did not detect any major undesired effect in the treated mice.

Referee #3 (Remarks for Author): The manuscript by Abir Arfaoui et al., entitled " A genome-wide RNAi screen reveals essential therapeutic targets of breast cancer stem cells" describes an integrated interactome/regulome analysis based on a genome-wide RNA interference screen in a cell-based model system of basal-type breast cancer, the SUM159 cell line. The ultimate goal of this study is the characterization of genes/pathways that might constitute suitable targets for a rationale anti-cancer stem cell (CSC) therapy in breast cancer. By this approach, the Authors identified potential underpinnings of CSC phenotypes, which were subsequently validated through in vitro and in vivo functional studies as suitable targets for pharmacological intervention using a number of available compounds. An important outcome of these analyses is the identification of at least two drugs, salinomycin, previously described as a selective anti-CSC drug, and JQ1, which can be used alone or in combination to target CSCs. The manuscript addresses important questions in the field of therapeutic management of breast cancer, first and foremost the necessity to target CSCs to achieve definitive tumor eradication, as apposed to currently available strategies that in most cases exert a predominant effect on the bulk tumor population. In this contention, a highlight of this work is the use of in vitro and in vivo functional assays in an attempt to dissect the effects of the selected compounds on the CSC subfraction, rather than on the bulk tumor population. This approach is in keeping with the emerging idea that functional assays that interrogate CSC biology should be routinely implemented in the evaluation of efficacy of anti-tumor drugs. Notwithstanding these important and valuable aspects, there are several pitfalls to this work that detrimentally affect its overall significance and clinical impact at this time.

General comment: One major point of concern is the use of the ALDH-based technique as the sole phenotypic assays to "count" CSCs throughout the study. Indeed, while the use of ALDH in combination with the basal-like SUM159 cell line appears to be a suitable strategy, entirely justified

for the execution of a high-throughput analysis, there is a sort of over-reliance on ALDH as a "universal" approach to measure CSCs across the different cell lines and patient-derived cells (the molecular subtypes of patient is not described) in the subsequent validation studies. At least in the case of basal-type breast cancer, other CSC markers can be also exploited (for instance the CD44/CD24 configuration) which might have aided a better definition of the effects of the compounds on the CSC population.

We thank the reviewer for his/her helpful comments.

The CD44+/CD24- phenotype has been utilized to isolate breast CSCs in different models of human primary tumor xenografts. However, its utility as CSC marker in vitro is somewhat limited by the observation that, frequently, a large percentage of cells within a cell line express these putative stem cell markers, irrespective of their CSC properties. For example, >95% of cells in basal BCLs (such as SUM159) display the CD44+/CD24- phenotype. Indeed, a previous study has demonstrated that the CD44+/CD24-phenotype does not isolate the tumorigenic population in these cell lines (Fillmore et al., Breast cancer research, 2008). Thus, evaluation of the CD44+/CD24- cell population in BCLs treated with different compounds will thus not reflect changes in the CSC population. Better than an additional CSC marker, the tumorsphere assay is a functional test used to evaluate the CSC content in a biomarker-independent manner. Using this assay, we confirmed, in vitro, the effect of drugs on the CSC population in BCLs. Moreover, the impact of the drug treatments on the CSC self-renewal was further proved in a secondary tumorsphere assay. These results are now reported in Figure 3H and Figure 4D.

Concerning the different patient-derived xenograft models used in this study we cannot detect any CD44+/CD24- population for CRCM404 and CRCM434, whereas after treatment no variation of the CD44+/CD24- cell population was observed for CRCM494 (ie 75% of CD44+/CD24- cells, data not shown). Nevertheless, we performed tumor re-implantation to functionally demonstrate that the variation of the ALDEFLUOR-positive cell population was correlated with a modification of the tumorigenic cell population

Notwithstanding this, the combined SUM159/ALDH approach revealed to be a suitable strategy to validate the effects of the identified genes on the putative ALDH-positive CSC fraction, considering the good correspondence between ALDH-positive cells and sphere-forming efficiency in vitro (at least in some experiments). In this regard, a major point of weakness is the absence of evidence that some of these genes are also functionally implicated in determining variations of the CSC pool in vivo. Such evidence would have strongly reinforced the significance of pharmacological studies in vivo, based on the rationale that those genes/pathways are the actual molecular target(s) for the selected drugs. Differently said, one would expect that the perturbation of the identified genes/pathways, singly or in combination, would recapitulate the effects of the pharmacological treatments, which would provide definitive preclinical proof-of-evidence of the suitability of using the candidate compounds as anti-CSC treatment(s) in the clinical setting.

We agree with the reviewer that it is crucial to establish the association between the subnetwork's genes/pathways, the pharmacological treatment effect, and the impact on the

CSC *in vivo*. Because, the final goal of this study is to develop a pharmacological approach to target breast CSCs, instead of testing shRNA constructs *in vivo* we proposed to evaluate gene expression profiling in PDXs (CRCM404 and CRCM434) treated with Salinomycin/JQ1 combination compared to their respective vehicle-treated tumors. Using metagene-based score calculated for each subnetworks (defined in Figure 2), we were able to show that only subnetwork-6 (mediator complex) and subnetwork-9 (Autophagy) presented a significant down-regulation in tumors treated with Salinomycin/JQ1 combination compared to the vehicle-treated tumors. These observations suggest that Salinomycin/JQ1 combination specifically targets their related subnetworks. These results are now reported in Figure 4M and Supplementary Figure 7C.

Another important point concerns the 80% cut-off for proliferative block/apoptosis effects used to identify candidate hits in the RNAi screen in the bulk SUM159 population. If the purpose of the study was to identify genes/pathways with a selective effect on CSCs, which are hold to be predominantly quiescent/slow-proliferating/apoptosis-resistant, rather than an effect of the bulk population of actively proliferating progenitors, it would have been more appropriate to evaluate the differential apoptosis/proliferative effects on the purified ALDH-positive and -negative subfractions. **The aim of the RNAi screen was to identify regulator genes modifying the proportion of bCSC.** We estimate that siRNA pools that induced a massive cell death (corresponding to less than 20% of cell survival compared to the control) would interfere with an accurate high content screening analysis, due to the scarcity of the remaining cells that introduced a high reproducibility bias in the CSC/non-CSC ratio computation. More than a biological choice, this cut-off was a technical decision. Still, it is worth mentioning that this cut-off excluded only 0.5% of the siRNA pools (93 out of 17,920 pools), most of them targeting essential genes for cell proliferation/survival, that wouldn't have been relevant as therapeutic targets. Exclusion of this negligible list of genes had a minimal impact on the analysis.

SUM159 cells are basal-type, as apparently are all the other cell lines utilized in the study, (while the molecular and clinical characteristics for the two patients appear not be reported). In this regard, if it is true that basal-type breast cancers are among the most biologically aggressive breast cancer subtypes, they also represent a minority of breast cancers (where the great majority is represented by luminal breast cancers which are also presumably driven by CSCs). Assessing the implication of the identified genes/pathways and related targeted therapies in other breast cancer subtypes would have greatly strengthened the overall biological and clinical impact of the study, in light of the emerging idea that breast CSCs might share molecular and phenotypical homologies regardless of the heterogeneity that distinguishes the different molecular subtypes at the bulk population level.

All our *in vitro* experiments have been performed on a panel of five breast cancer cell lines (BCLs) representing the breast cancer molecular diversity (1 luminal BCL, 2 Basal BCLs, and 2 Mesenchymal BCLs). Similar results have been obtained for all these BCLs. For our *in vivo* experiments, we have focused our study on the triple-negative breast cancers (TNBCs). Even if TNBCs represent only 15-20% of the breast cancers, they present the worse prognosis with a lack of targeted therapies. We have now added the molecular characteristics of the different

cell line and PDX models in the method section. Concerning the clinical impact of our subnetwork expression, the analysis has been done on a cohort of 2578 patients representing the breast cancer inter-tumoral heterogeneity.

The putative anti-CSC drugs identified, salinomycin - which has been already described as an anti-CSC drug -, JQ1 - which should target the sub-network mediator complex - and mifeprestone - that targets NR3C1- appear to show highly variable, sometimes not reproducible, effects when their efficacy in vitro (ALDH/sphere-forming assay) is compared to their efficacy in vivo. Moreover, only salinomycin shows a consistent effect when used alone or in combinatorial treatment, while all the other drugs display a modest, if any, effect when used alone.

Furthermore, accurate evaluation of the growth kinetics of xenografts in vivo appears to show a marked anti-proliferative effect in the bulk tumor population. This point should be carefully addressed with an accurate analysis of the rate of apoptosis and proliferation in the treated bulk xenograft population. For instance (Fig. 5), the effects of salinomycin and JQ1 on the CSC content, and on the growth rate of the primary xenograft appear to be inconsistent compared to each other. In addition, the absence of effect of salinomycin on the growth of the primary xenograft (which is expected of a selective anti-CSC therapy) does not correspond to a consistent action of salinomycin in reducing the content of ALDH-positive CSCs; even more concerning, JQ1 appears to be endowed with a remarkable anti-tumor growth activity (which is of the same order of magnitude of the combined salinomycin+JQ1 treatment) which is not accompanied by any substantial effect of this drug on the ALDH-positive fraction. This suggests that the site of action of JQ1 might be at the level of the bulk population of progenitors, rather than hitting the CSC population. Again, while the effects of the different treatments on the growth of the secondary xenografts are remarkable, a thorough analysis of the apoptosis/proliferative effects in the bulk tumor population would aid understanding of the mechanism/site of action of the different treatments. Therefore, in light of the different caveats and pitfalls of the above in vivo studies, the conclusion that these drugs represent selective anti-CSC treatments remains questionable.

We agree with the reviewer that changes in bCSC proportion may be explained by different scenarios such as, induction of apoptosis in the CSC subpopulation, modification of proliferation rate of one or both cell subpopulation(s), CSC differentiation, or a combination of those. Depending on these scenarios the growth rate of the treated tumors may be different with a similar impact on the bCSC proportion. To evaluate how drugs affect bCSC proportion we have first measured the proportion of apoptotic cells in each cell subpopulations following treatments (JQ1, SAL, JQ1/SAL). We did not observed a significant increase of apoptotic cells in each cell subpopulations following drug treatments. These results are now reported in Figure 4E. To evaluate the impact of the treatment on the cell proliferation and on the bCSC differentiation potential we developed a lineage tracing system using an engineered SUM 159 cell line (see new Figure 4F). Using this model we can follow the non-CSC (ALDH^{neg}-RFP⁺ cells) and CSC populations (ALDH^{pos}-BFP⁺ cells) progenies under treatment conditions. We did not observed any modifications of the RFP⁺/BFP⁺ cells ratio in the treated conditions

compared to the control, indicating that treatments did not induce a differential proliferation rate of each cell subpopulations. However, we did observe a significant reduction of the ALDHbr cell proportion in the CSC progenies (BFP+ cells) under treatments compared to the control (see new Figure 4G-I). These results suggest that JQ1 and Sal treatments mainly reduced the CSC proportion by promoting CSC differentiation.

In the light of these last observations, *in vivo* data may be reinterpreted. We performed a KI67 staining on tissue sections from treated tumors. We did not observe any differences in term of cell proliferation between tumors from the different treatment conditions compared to the control. We also performed a cleaved-caspase-3 staining to evaluate the proportion of apoptotic cells. In JQ1-treated and Sal/JQ1-treated tumors we observed a significant increase of the proportion of apoptotic cells explaining the anti-tumor growth activity, whereas we did not observed any increase of the apoptotic cell proportion in the salinomycin-treated tumors compared to the control. These results are in line with a differentiation therapy effect in the former group (JQ1 and Sal/Jq1) that is often associated with an induction of apoptosis in more mature cells. It is also concordant with the limited effect of salinomycin treatment on PDXs growth accompanied by a decrease of CSC proportion. All these results are now reported in Supplementary Figure 7B-C.

Major points:

Fig. 3. It would be useful to show, in addition to the sphere-forming efficiency in panel B, a table reporting the actual average size of the spheres (expressed as number of cells/sphere) alongside measurement of the actual proliferative rate (for instance, cell cycle analysis; also in panel C/D) in the bulk sphere population, with the ultimate goal to demonstrate the differential effects of the treatments on the putative CSC (ALDH-pos) vs. progenitor (ALDH-neg) population. A serial propagation of spheres (at least a second generation) would definitively aid addressing the selective anti-CSC effect of the various treatments, at least for selected genes/drugs.

The effect of mifepristone and JQ1, shown in Fig. G is really modest, compared to salinomycin, which questions their actual efficacy in this assay. As previously stated, also in this analysis, the analysis of the actual sphere size and of the apoptosis/proliferative rate would probably be instrumental to better highlighting the intrinsic mechanism/cellular site of action of these compounds.

As answered in the previous comment, we have now determined the functional mechanism of action of JQ1 and Sal on the CSC proportion. Both compounds induced the differentiation of the CSC population. As suggested by the reviewer, to further evaluate the impact of each treatment on the CSC self-renewal, we have performed secondary tumorsphere assay. We confirmed our observed results in primary tumorsphere assay with a significant reduction of the secondary tumorsphere-forming cell proportion following JQ1, Sal, or the drug combination treatments. These results are now reported in Figure 3H and Figure 4D.

Fig. 4. Results depicted in this figure would probably be better represented by a synoptic table showing the selected final optimal concentrations of the different drugs, their effects in terms of sphere-forming efficiency and the intrinsic characteristics of the different cell lines (subtypes).

To clarify this point we have now mentioned in the method section the different drug concentrations used to perform the tumorsphere assays.

The experiments depicted in panel C lacks fundamental controls (not-treated control, salinomycin alone at 30 and 500nM) which are required to understand whether this drug is already effective as a standalone treatment, or whether a combinatorial regimen is required to maximize its efficacy, especially in light of the relatively modest effects that JQ1, and to some extent also salinomycin, appear to exert when used in mono-therapy. This would also aid understanding whether the different drugs are hitting different CSC pools (or different molecular workings), considering their synergic effects when used in combination, or whether their combination simply results in off-targets effects (toxicity?).

We have followed the reviewer's recommendations and we have now added flow charts for each missing controls (see new Figure 4B)

In these experiments, it also concerning that the remarkable reduction (~8-10-fold) observed in the percentage of ALDH-positive cells upon inhibitor treatments (panel C) does not reflect into an equivalent effect in the sphere-forming assay. Once again, accurate dissection of the possible effects on proliferation/apoptosis in these assays, accompanied by second generation sphere formation assays, would help addressing these points. It is also possible that ALDH is not sufficient to "read" the selective effects on the CSC population, which might be better interpreted with different approaches to CSC purification (i.e. CD24/CD44-based sorting, if amenable).

We agree with the reviewer that the effect of drug treatment on the ALDH^{br} cell population does not translate into an equivalent effect in the tumorsphere assay. The main reason comes from the ALDH phenotype that defines heterogeneous cell populations. The ALDH^{br} cell population is highly enriched in CSC compared to the ALDH⁻ cell population but it doesn't exclude that some CSC are contained in the ALDH⁻ cell population. Actually, we previously published that even though ALDH^{br} cell from different cell line models are the most efficient subpopulations at generating tumorspheres, the ALDH⁻ cell subpopulations are also able to generate some tumorspheres, although to a much lesser extent (see Charafe-Jauffret et al., Cancer Research, 2009). To further validate the impact of the treatment on the CSC population we have now performed secondary tumorsphere-formation assay. These new experiments confirmed the primary tumorsphere-formation assay with significant decrease of the number of secondary tumorsphere-formed in the drug treated conditions (see new Figure 4D).

Fig. 5. This figure depicts results of possible translational relevance, and as such represents a major added value of the study. Increasing the number of patients including different breast cancer subtypes would greatly increase the impact of these results, also considering the remarkable

variations observed in the analysis of the two patients. Are these patients p53 wild-type or mutated? More in general, what are the intrinsic characteristics of these two patients?

Concerning our in vivo experiments, we have focused our study on the triple-negative breast cancers that are still facing unmet medical needs. To reinforce our study, we have now tested one additional PDX model of triple-negative breast cancer. We obtained similar results in all three PDX models tested with a significant effect of the salinomycin/JQ1 drug combination on the CSC population, limiting tumor progression. These results are now reported in Figure 4 and in Supplementary Figure 6. The description of our PDX models has been added in the Material and Methods section. Of note, all our PDX models are p53 mutated as 80% of the triple-negative breast cancers.

A thorough analysis of the effects of the drug treatments (proliferation/apoptosis) is also required, as already suggested above. It is also not very well described when the treatments were started. Again, as previously stated, salinomycin and JQ1 appear to have variable, inconsistent effects on the CSC content, when the ALDH method is compared to the sphere-forming assay.

As answered previously, we have now measured the proliferation and apoptotic rates in all the drug treated conditions compared to the control (see comments above). For better clarity, we mentioned on each growth curves (Figure 4B, Supplementary Figure 6A, 6E), the starting time of the drug treatments. The overall treatment period was materialized by a grey area.

In the limiting dilution transplantation assays, while some treatments appear to have a more remarkable effect than others, the combinatorial treatments result in a dramatic inhibition of the tumor growth, which raises concerns about the selective anti-CSC effects of these treatments. This point should be carefully addressed by corroborating these studies with functional (sphere-forming efficiency, ALDH assay) and in situ (detection of apoptosis/proliferation rate) analyses of the tumor outgrowths. Taken at a face value, results from in vivo studies appear to indicate that a combinatorial treatment is not strictly required to achieve a substantial effect.

The reimplantation assay using limiting dilution is the current gold standard to evaluate residual CSC in a treated tumor. Using this assay we clearly demonstrated that the drug combination presents a better potential to reduce the CSC pool compared to the drugs used as single agents. Moreover, we have now performed a metastasis-formation assay, demonstrating the superior efficiency of the drug combination compared to single agent treatments in reducing the metastatic burden.

Towards assessing the preclinical relevance of these studies, it would also be useful to evaluate the combination of (at least one) candidate drugs with standard chemotherapy, to observe whether the selective anti-CSC action of one of these compounds prevent tumor growth relapse after chemotherapy discontinuation (as it would be expected in case of intrinsic CSC resistance to chemotherapy).

We agree with the reviewer that it is important to associate our drug combinations with standard chemotherapy. We have now tested this association (using docetaxel as standard

chemotherapy), *in vitro*, in two different cell line models. We obtained similar results for both models with JQ1, Sal, or JQ1/Sal treatments preventing CSC selection induced by docetaxel treatment. These results are now reported in Supplementary Figure 4. The next step will be to develop a preclinical assay in PDXs testing the different regimens (concomitant, sequential) of tri-therapies (JQ1/Sal/Docetaxel). This step will be mandatory to translate our therapeutic approach in clinics. However, we think that it is beyond the scope of the present manuscript.

Additional points:

In some parts, the manuscript does not read well. The definition "breast CSC" or the acronym bCSC are inconsistently used throughout the manuscript.

We thank the reviewer and made the correction.

2nd Editorial Decision

10th July 2019

Thank you for the submission of your revised manuscript to EMBO Molecular Medicine, and please accept my apologies for the delay in getting back to you, which is due to the fact that one referee needed more time to complete his/her report. We have now heard back from the three referees who were asked to reassess your work. While referees 1 and 2 are now supportive of publication, referee 3 still expresses concerns that we would like you to address in writing before we can accept your manuscript for publication.

1) Referees' comments:

Please address all referee 3's comments in writing. At this stage, we'd like you to discuss this referee's concerns, and if you do have data at hand, we'd be happy for you to include it, however we will not ask you to provide any additional experiments.

REFeree REPORTS

Referee #1 (Remarks for Author):

The authors addressed all the comments I had appropriately.
I have no further comment

Referee #2 (Comments on Novelty/Model System for Author):

The manuscript has undergone an excellent revision and of very good technical quality. The novelty is high.

Referee #2 (Remarks for Author):

The authors have addressed all of my major and minor concerns. The manuscript is of excellent quality and should benefit general readership and specialist.

Referee #3 (Remarks for Author):

I believe that, despite the attitude to use some sort of circular arguments to circumvent experimental

hurdles (for instance, it is not clear why the CD44/CD24 configuration does not work in triple-negative human breast cancer xenografts, nor is it really convincing that the ALDH-based approach is less plagued than the CD44/CD24 configuration in detecting mixed heterogeneous stem+progenitor cells populations!), the Authors provided a number of functional in vivo and in vitro assays that addressed, at least in part, some of the original concerns of this Reviewer on the functional relevance of the findings of this study (first and foremost, limiting dilution transplantation and metastatic potential studies, and secondary tumorsphere assay).

However, some concerns and major critical points still remain, which detrimentally affect the overall strength of the study. For instance, as to the functional validation of selected subnetwork's genes/pathways in response to pharmacological treatments, I believe that the gene expression profiling approach, that was used as an alternative to shRNA-based genetic studies, is largely far from being an acceptable approach to establish mechanistic associations, as it does only allow to infer the targeted regulations of some of those subnetworks/genes by the candidate anti-cancer stem cell treatments. Although this Reviewer is aware of the amount of work required for these experiments, the availability of considerable amount of human PDX material and suitable technology to the Authors' lab makes these experiments entirely feasible in a reasonable time-frame.

It also detrimental to the overall impact of the study, the decision to focus on triple-negative human PDXs and overlook the possible relevance of the candidate treatments to luminal breast cancers. Indeed, while luminal tumors are the great majority of human breast cancers, most surprisingly, the Authors appear to misrecognize that, if it is true that triple-negative breast tumors represent poor prognosis disease with substantial lack of targeted therapies, a major unmet clinical need in breast cancer is also represented by the absence of targeted anti-cancer stem cell therapies to prevent relapse in luminal breast cancers, in which there is a persistent 1-2% yearly rate of distant recurrences, with more that 50% of relapses occurring after 5 years after the initial diagnosis (which clearly point to the functional implication of long quiescent tumorigenic cells endowed with stemness traits).

The SUM159-based lineage tracing systems does not experimentally address the point of the specificity of the different treatments in targeting strict sense cancer stem cells vs. progenitors, as the ALDH phenotype is by no means a selective cancer stem cell tracer, and rather labels heterogeneous cell populations, as also explicitly agreed on by the same Authors in their point-by-point-reply. The persistent discrepancy between tumorsphere assay and ALDH phenotype remains a major controversial point of this study.

I do believe that the entire emphasis on the targeted action of the drugs analyzed in this study, in the absence of mechanistic studies, and on the selective targetability of cancer stem cells should be overall heavily downtoned.

2nd Revision - authors' response

13th July 2019

Referee #1 (Remarks for Author):

The authors addressed all the comments I had appropriately. I have no further comment

We thank the reviewer for his/her helpful comments during the reviewing process.

Referee #2 (Comments on Novelty/Model System for Author):

The manuscript has undergone an excellent revision and of very good technical quality. The novelty is high.

Referee #2 (Remarks for Author):

The authors have adressed all of my major and minor concerns. The manscript is of excellent quality and should benefit general readership and specialist.

We thank the reviewer for his/her helpful comments during the reviewing process.

Referee #3 (Remarks for Author):

I believe that, despite the attitude to use some sort of circular arguments to circumvent experimental hurdles (for instance, it is not clear why the CD44/CD24 configuration does not work in triple-negative human breast cancer xenografts, nor is it really convincing that the ALDH-based approach is less plagued than the CD44/CD24 configuration in detecting mixed heterogeneous stem+progenitor cells populations!), the Authors provided a number of functional in vivo and in vitro assays that addressed, at least in part, some of the original concerns of this Reviewer on the functional relevance of the findings of this study (first and foremost, limiting dilution transplantation and metastatic potential studies, and secondary tumorsphere assay).

However, some concerns and major critical points still remain, which detrimentally affect the overall strength of the study. For instance, as to the functional validation of selected subnetwork's genes/pathways in response to pharmacological treatments, I believe that the gene expression profiling approach, that was used as an alternative to shRNA-based genetic studies, is largely far from being an acceptable approach to establish mechanistic associations, as it does only allow to infer the targeted regulations of some of those subnetworks/genes by the candidate anti-cancer stem cell treatments. Although this Reviewer is aware of the amount of work required for these experiments, the availability of considerable amount of human PDX material and suitable technology to the Authors' lab makes these experiments entirely feasible in a reasonable time-frame.

We thank the reviewer for his/her helpful comments

We agree with the reviewer that an shRNA-based genetic study in PDXs would have been an informative approach to decipher the underlying molecular mechanisms. However, besides the considerable amount of additional work required (lentiviral constructs design/production for different target genes, PDXs infection in different models, PDXs growth monitoring, and reimplantation assays), our final goal was to develop a pharmacological approach to target breast CSCs. Thus, we decided to focus our effort on drug testing in order to provide a preclinical proof-of-evidence.

It also detrimental to the overall impact of the study, the decision to focus on triple-negative human PDXs and overlook the possible relevance of the candidate treatments to luminal breast cancers. Indeed, while luminal tumors are the great majority of human breast cancers, most surprisingly, the Authors appear to misrecognize that, if it is true that triple-negative breast tumors represent poor prognosis disease with substantial lack of targeted therapies, a major unmet clinical need in breast cancer is also represented by the absence of targeted anti-cancer stem cell therapies to prevent relapse in luminal breast cancers, in which there is a persistent 1-2% yearly rate of distant recurrences, with more that 50% of relapses occurring after 5 years after the initial diagnosis (which clearly point to the functional implication of long quiescent tumorigenic cells endowed with stemness traits).

We do not misrecognize that hormone-resistant luminal tumors present an unmet clinical need. However it would have been unrealistic to perform the drug testing in different panels of PDXs covering all the different molecular subtypes. This study constitutes a first step in the identification of new potential therapeutic approaches to target the bCSCs in triple-negative breast cancers. We agree that further experiments are needed to extend these observations to others molecular subtypes (Luminal, ERBB2 ...). However, this work lies beyond the scope of the present manuscript.

The SUM159-based lineage tracing systems does not experimentally address the point of the specificity of the different treatments in targeting strict sense cancer stem cells vs. progenitors, as the ALDH phenotype is by no means a selective cancer stem cell tracer, and rather labels heterogeneous cell populations, as also explicitly agreed on by the same Authors in their point-by-point-reply. The persistent discrepancy between tumorsphere assay and ALDH phenotype remains a major controversial point of this study.

I do believe that the entire emphasis on the targeted action of the drugs analyzed in this study, in the absence of mechanistic studies, and on the selective targetability of cancer stem cells should be overall heavily downtoned.

As mentioned in our first point-by-point-reply, the apparent discrepancies between ALDH phenotypes and tumorsphere assays can be explained by the transient capacity of some ALDH-negative cells to generate tumorsphere. Moreover, tumorspheres are also known to be enriched in CSCs and progenitors cells rather than a pure CSC population. It is commonly accepted that each tumorspheres passage challenged the *in vitro* self-renewal potential of CSCs. Of note, JQ1/Salino treatment decrease the primary tumorsphere formation of 65% compared to the control whereas it decreases the secondary tumorsphere formation of 88% compared to the untreated conditions. These observations are clearly in favor of an effect of the JQ1/Salino treatment on the reduction of the CSC population.

That being said, we agree with the reviewer that the “selective targetability” of the CSCs is not unequivocally demonstrated in our study. Thus, we have modified the text as follow:

“These results support the use of this drug panel as selective inhibitors of the bCSC-fate.”

is replaced by:

“These results support the use of this drug panel as potential inhibitors of the bCSC-fate.”

“Altogether, these results support that a combination of JQ1 and salinomycin is a promising therapeutic approach to selectively target the bCSC population.”

is replaced by:

“Altogether, these results support that a combination of JQ1 and salinomycin is a promising therapeutic approach to reduce the bCSC proportion.”

“Figure 4. Drug combination screen identifies salinomycin as acting synergistically with JQ1 to selectively target the bCSC population.”

is replaced by:

“Figure 4. Drug combination screen identifies salinomycin as acting synergistically with JQ1 to reduce the bCSC proportion.”

YOU MUST COMPLETE ALL CELLS WITH A PINK BACKGROUND ↓
PLEASE NOTE THAT THIS CHECKLIST WILL BE PUBLISHED ALONGSIDE YOUR PAPER

Corresponding Author Name: GINESTIER Christophe
Journal Submitted to: EMBO Molecular Medicine
Manuscript Number: EMM-2018-09930